 **eLIFE**

# Regulation of EGFR signal transduction by analogue-to-digital conversion in endosomes

**Roberto Villaseñor[1], Hidenori Nonaka[1], Perla Del Conte-Zerial[1], Yannis Kalaidzidis[1,2], Marino Zerial[1]\***

[1]Max Planck Institute of Molecular Cell Biology and Genetics, Dresden, Germany; [2]Faculty of Bioengineering and Bioinformatics, Moscow State University, Moscow, Russia

**Abstract** An outstanding question is how receptor tyrosine kinases (RTKs) determine different cell-fate decisions despite sharing the same signalling cascades. Here, we uncovered an unexpected mechanism of RTK trafficking in this process. By quantitative high-resolution FRET microscopy, we found that phosphorylated epidermal growth factor receptor (p-EGFR) is not randomly distributed but packaged at constant mean amounts in endosomes. Cells respond to higher EGF concentrations by increasing the number of endosomes but keeping the mean p-EGFR content per endosome almost constant. By mathematical modelling, we found that this mechanism confers both robustness and regulation to signalling output. Different growth factors caused specific changes in endosome number and size in various cell systems and changing the distribution of p-EGFR between endosomes was sufficient to reprogram cell-fate decision upon EGF stimulation. We propose that the packaging of p-RTKs in endosomes is a general mechanism to ensure the fidelity and specificity of the signalling response.

**\*For correspondence:** zerial@mpi-cbg.de

**Competing interests:** The authors declare that no competing interests exist.

## Introduction

Cells respond to various signals by activating different types of RTKs and committing to specific cell-fate decisions (*Katz et al., 2007*). A remarkable property of this system is that different RTKs can elicit distinct cellular responses through the same signal transduction machinery (*Marshall, 1995*; *Kholodenko et al., 2006*). In several cases, signalling specificity results from differences in amplitude and duration of the intracellular signalling cascades (*Marshall, 1995*; *Maroun et al., 2000*; *Nagashima et al., 2007*). For example, in PC12 cells, EGF stimulation of EGFR leads to transient Erk phosphorylation and cell proliferation, whereas NGF binding to TrkA leads to sustained Erk phosphorylation and cell differentiation (*Marshall, 1995*). Differences in signalling amplitude and duration can arise from positive or negative feedback loops within the same signalling pathway (*Santos et al., 2007*) or activation of additional signalling components (*York et al., 1998*). To explain such differences, it has been proposed that both EGF and NGF stimulation induce a specific 'molecular context' that determines the topology of the signal transduction network (*Santos et al., 2007*). How such a topology is determined for different RTKs and whether it is the sole determinant of signal specificity is unclear (*Kholodenko, 2007*).

Insights into this problem may be provided by the spatio-temporal distribution of RTKs along the endosomal system. The detection of phosphorylated receptors and signalling adaptors in endosomes (*Di Guglielmo et al., 1994*; *Vieira et al., 1996*; *Sorkin, 2001*; *Teis et al., 2002*; *Lampugnani et al., 2006*; *Galperin and Sorkin, 2008*; *Schenck et al., 2008*; *Coumailleau et al., 2009*) led to the concept that signalling is initiated at the plasma membrane but continues in endosomes (*Di Guglielmo et al., 1994*).

**eLife digest** Molecules called growth factors can stimulate cells to grow, divide, or differentiate into more specialised cell types. Cells detect these molecules via proteins called receptor tyrosine kinases that span their surface membrane. The growth factor binds to the portion of the receptor outside the cell, which makes the receptor send signals to the cell's nucleus that change how the cell grows, divides, or specialises.

Different growth factors and receptor tyrosine kinases affect cell development in different ways. However, it was unclear how this occurred, as the receptors all send signals via the same signalling pathways. Some researchers proposed that specific responses could be triggered if some receptor tyrosine kinases activated these pathways more strongly than other receptors, or if they activated the pathways for different lengths of time.

Now, Villaseñor et al. have looked at a receptor tyrosine kinase for a growth factor called EGF. Activated EGF receptors are marked with a phosphate group (or 'phosphorylated') and are then removed from the surface membrane and packaged into structures within the cell called endosomes. Villaseñor et al. found that different endosomes contain the same mean amount of phosphorylated EGF receptor. When exposed to higher EGF concentrations, the cells respond by increasing the number of endosomes, and so the average number of phosphorylated EGF receptors in each endosome remains almost constant. Villaseñor et al. used mathematical modelling to show that this mechanism, which they refer to as an 'analogue-to-digital conversion', ensures a robust signal, and can regulate the signalling of the activated receptors in both space and time.

Different growth factors either increase or decrease the number and size of endosomes in various cell types. Moreover, Villaseñor et al. found that changing how the phosphorylated EGF receptor is distributed between endosomes alters how the cells interpret the signal and differentiate when they are exposed to EGF. These findings mean that the signalling activity of a cell could be predicted from the number and size of its endosomes. Moreover, the findings also suggest that interfering with this mechanism could change cell behaviour, for example, it could stop cancer cells proliferating and force them to differentiate instead.

Indeed, inhibition of endocytosis by blocking Dynamin function causes significant alterations in signalling specificity (*Vieira et al., 1996*). However, recent studies challenged this concept arguing that EGFR signalling occurs primarily at the plasma membrane (*Damke et al., 1994*; *Brankatschk et al., 2012*; *Sousa et al., 2012*). Interestingly, a recent systems survey of endocytosis (*Collinet et al., 2010*) revealed an unexpected tight control in the number, size, and cargo content for EGF-positive endosomes, raising the question of why is EGF packaging in endosomes so accurately controlled? Here, we hypothesized that the tight control of the endosomal distribution of EGF could serve to regulate signal transmission. We tested this hypothesis by quantitatively analysing the endosomal distribution of EGFR as an RTK model system in endosomes and evaluating its impact on cell-fate decisions.

## Results

To measure the content of p-EGFR in individual endosomes, we used two independent assays (for a detailed description see 'Materials and methods' and *Figure 1—figure supplement 1*). First, we modified a FRET-FLIM microscopy assay previously used to measure the spatial distribution of p-EGFR at the plasma membrane (*Wouters and Bastiaens, 1999*; *Verveer et al., 2000*). The assay measured the FRET signal between EGFR-GFP and an anti-phospho-tyrosine antibody (p-Tyr-ab) labelled with AlexaFluor 555. Since FLIM microscopy lacks the spatial resolution to analyse the receptor activation at a sub-cellular level, we modified the assay into a high-resolution FRET microscopy assay. However, instead of the total cell signal, we measured the distribution of EGFR and p-EGFR at the level of individual endosomes resolved by high-resolution confocal microscopy and quantitative automated image analysis (*Rink et al., 2005*; *Collinet et al., 2010*) (*Figure 1—figure supplement 2*). To avoid artefacts of overexpression, we used HeLa cells transfected with a bacterial artificial chromosome (BAC) transgene stably expressing EGFR-GFP under its endogenous promoter (*Poser et al., 2008*). In these cells (*Figure 1—figure supplement 3A*), the uptake of EGF was only ~twofold higher compared to

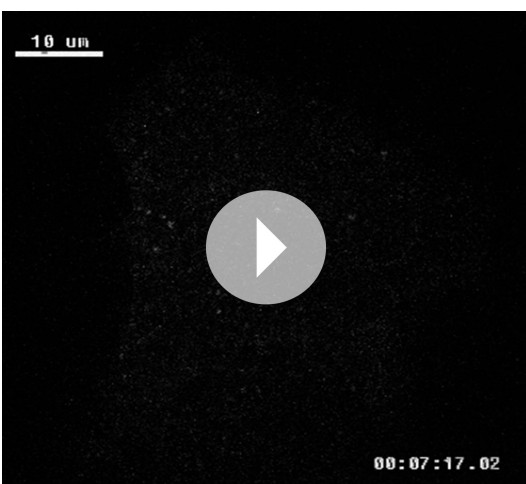

**Video 1.** Live-cell imaging of EGFR endocytosis. HeLa EGFR-GFP BAC cells were imaged with a spinning disk microscope after 1 minute of EGF stimulation with 10 ng/ml EGF. Movie shows maximal projection of 3 z-slices of 0.8 mm thickness.

endogenous (*Figure 1—figure supplement 3B*). However, the transport kinetics were similar (*Figure 1—figure supplement 3C*). Second, we measured p-EGFR with an anti-body against a specific phospho-tyrosine residue (Tyr1068). Both assays gave very similar results (*Figure 1—figure supplement 4*). As the FRET assay is not restricted to a single phosphorylation site that can change over time (*Morandell et al., 2008*), we used it as a primary assay in further experiments. Under the fixation conditions used, we observed no significant difference in the morphology (*Figure 1—figure supplement 5A*) or area (*Figure 1—figure supplement 5B*) of EGFR-positive endosomes (*Video 1*). For every time point, ~15,000 endosomes from over 200 cells were analysed.

Continuous stimulation with EGF triggered the internalization of EGFR into endosomes (*Figure 1* and *Figure 1—figure supplement 2*). The total amount of endosomal EGFR peaked after 15 min and decreased, reflecting (1) down-regulation of surface receptors (*Wiley et al., 1991*) and (2) their degradation (*Dunn and Hubbard, 1984*) over time (*Figure 1A*, green curve). On the other hand, the total p-EGFR levels reached a maximum already at 10 min, followed by a phase of decay (*Figure 1A*, red curve). Comparison of decay kinetics for both curves after 15 min showed that de-phosphorylation of p-EGFR occurred faster than degradation ($\tau_{\text{decay EGFR}} = 88.13 \pm 14.49$, $\tau_{\text{decay p-EGFR}} = 30.97 \pm 1.69$, for details see 'Materials and methods'). Our FRET measurements are thus consistent with previously reported EGFR transport and phosphorylation kinetics determined by biochemical and microscopic methods (*Di Guglielmo et al., 1994*; *Burke et al., 2001*).

We next determined the distribution of EGFR and p-EGFR in individual endosomes. The number of endosomes with p-EGFR decayed with similar kinetics as the total p-EGFR signal ($\tau_{\text{decay N-p-EGFR}} = 45.24 \pm 11.39$ vs $\tau_{\text{decay p-EGFR}} = 30.97 \pm 1.69$; compare red with black curve in *Figure 1—figure supplement 6*). The mean content of total EGFR per endosome increased over time and then rapidly decayed reaching steady state, due to the balance of continuous EGF uptake and degradation (*Figure 1B*, green curve). After a rapid increase, the mean content of p-EGFR in each endosome stabilized to a fairly constant level after ~20 min (*Figure 1B*, red curve). Similar results were obtained when EGF was pulsed for 1 min and chased for different periods of time (*Figure 1B*, blue and black points).

To determine how the endosomal content of p-EGFR originates over time, we compared the distributions of EGFR and p-EGFR content per endosome. The width of distribution of total EGFR increased with time (*Figure 1C*), due to the fact that, as EGF continues to flow in, it first enters small early endosomes and progressively accumulates in larger ones (*Rink et al., 2005*). In contrast, the p-EGFR distribution first widened like that of total EGFR but then became almost twofold narrower than that of EGFR (*Figure 1—figure supplement 7A–E* compare the red and green curves) and stabilized after 30 min (*Figure 1D*). These results suggest an unexpected behaviour of p-EGFR, which over time stabilizes at a constant mean level per endosome.

Surprisingly, the mean amount of p-EGFR in endosomes was not ligand dependent. We stimulated cells with different concentrations of EGF for 30 min, when the amount of p-EGFR per endosome reached its steady state (*Figure 1B,D*). Once again, we found that EGFR and p-EGFR behaved very differently. The number of endosomes containing EGFR saturated already at low concentrations of EGF (0.5–1.0 ng EGF; *Figure 1E*, green curve) whereas the amount of total EGFR per endosome increased almost linearly (*Figure 1F*, green curve). This is expected because the higher the concentration of EGF, the higher the internalization of EGFR, whereas the number of receiving endosomes does not change significantly. In contrast, the number of endosomes with p-EGFR augmented with increasing EGF concentrations (*Figure 1E*, see red curve in

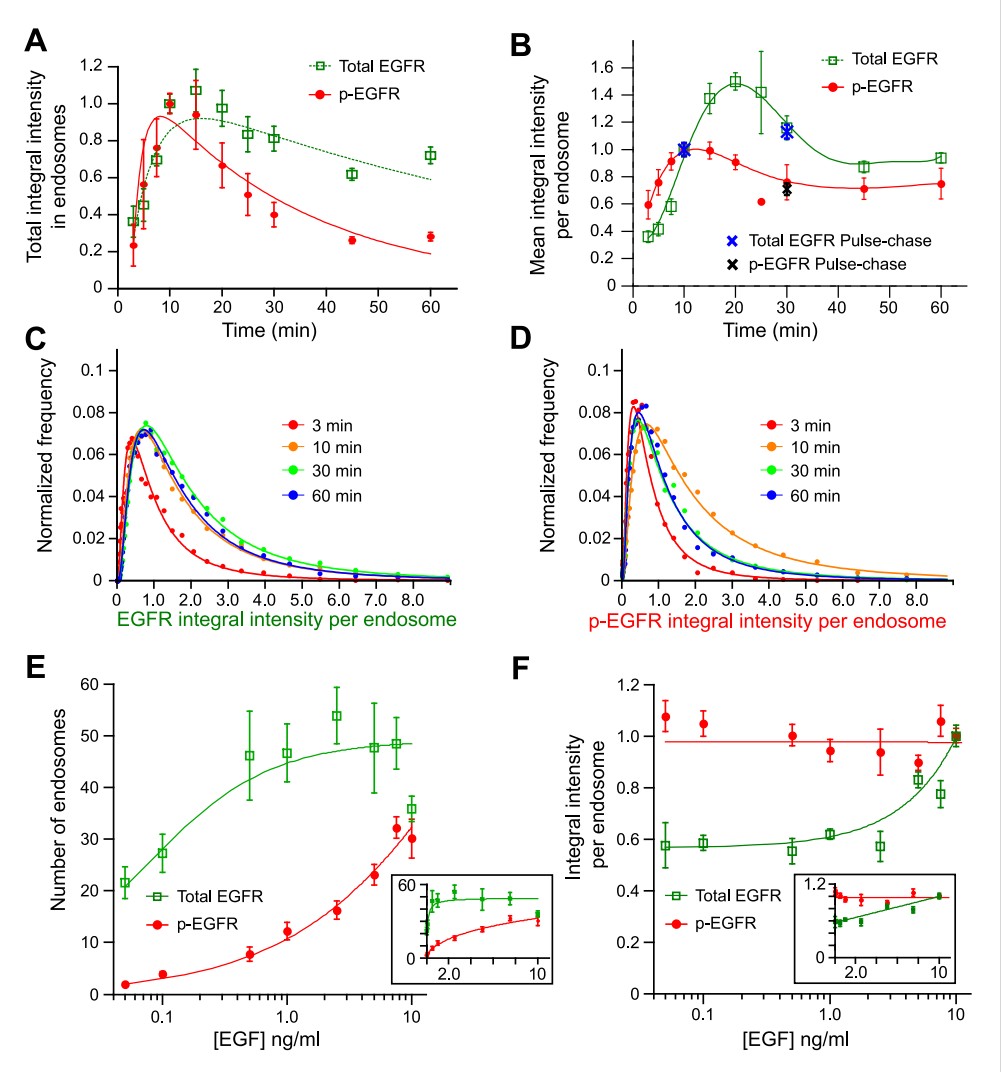

**Figure 1**. Cells keep a constant amount of p-EGFR in endosomes. (**A**) Time course of total integral intensity of EGFR (green) and p-EGFR (red) in endosomes measured by a FRET microscopy assay in HeLa EGFR BAC cells after continuous stimulation with 10 ng/ml EGF. The total integral intensity is defined as the sum of integral intensities of all endosomes in an image normalized by the area covered by the cells (for details see 'Materials and methods' and Supplementary information). (**B**) Time course of mean integral intensity per endosome for total EGFR (green curve) and p-EGFR (red curve) as in (**A**). Intensity curves (**A–B**) were normalized to the intensity value at 10 min. Crosses show the corresponding values after 1 min of EGF stimulation and incubation in ligand-free medium for 10 or 30 min (pulse-chase). (**C**) Time course of histogram distributions of the total EGFR integral intensity per endosome upon EGF stimulation as in (**A**). (**D**) Time course of histogram distributions of the p-EGFR integral intensity per endosome upon EGF stimulation as in (**A**). In both graphs, receptors in CCVs are responsible for the width of the distribution at 3 min (red curves in **C** and **D**). For comparison, histogram amplitude in **B** and **C** were normalized by each curve integral. In each graph, the integral intensity values were scaled by the mode of the histogram at 10 min. The experimental points from all histograms were fitted with a log-normal distribution. (**E–F**) Distribution of p-EGFR in endosomes as a function of EGF concentration after continuous stimulation for 30 min. Mean number of endosomes with EGFR (green curve) and p-EGFR (red curve) per 1000 µm² of the area covered by cells (**E**) and mean integral intensity of EGFR (green curve) and p-EGFR (red curve) per endosome (**F**). On panel (**F**) curves were normalized to the intensity value at 10 ng/ml EGF. Lines are hyperbolic fits (**E**) or least square fits (**F**) to the experimental points. In both cases insets show the same graphs in linear scale. The different magnitude of the error bars in (**E**) and (**F**) is due to the averaging by the total number of images (**E**) or the total number of endosomes (**F**). In all cases, points show mean ± SEM. All measurements were done in three independent experiments with a total of ~150 cells per time point or condition.

The following figure supplements are available for figure 1:

**Figure supplement 1**. Bleed-through correction for p-EGFR detection by FRET microscopy.

*Figure 1. continued on next page*

*Figure 1. Continued*

**Figure supplement 2**. EGFR and p-EGFR measurements by FRET microscopy.

**Figure supplement 3**. BAC expression of EGFR-GFP does not change EGF transport kinetics.

**Figure supplement 4**. Validation of FRET measurements with a specific anti-Tyr1068 antibody.

**Figure supplement 5**. PFA fixation does not significantly change endosome EGFR endosome morphology.

**Figure supplement 6**. The total amount of p-EGFR in endosomes decays with the same kinetics as the number of endosomes with p-EGFR.

**Figure supplement 7**. p-EGFR has a narrower integral intensity per endosome distribution than the total EGFR at late time points.

**Figure supplement 8**. The mean amount of p-EGFR per endosome increases at high concentrations of EGF.

**Figure supplement 9**. The mean p-EGFR amount per endosome does not correlate with endosome area at late time points after EGF stimulation.

semi-logarithmic scale, inset in linear scale). Strikingly, the mean amount of p-EGFR per endosome remained fairly constant, despite the EGF concentration varying almost over three orders of magnitude (*Figure 1F*, red curve). Therefore, increasing concentrations of EGF resulted in an increase in the number of endosomes with the same mean package of p-EGFR. Importantly, such a packaging is saturable because at high EGF concentrations the mean p-EGFR content per endosome was no longer constant with time (*Figure 1—figure supplement 8*).

The finding that endosomes contain a constant mean level of p-EGFR is striking. We performed several control experiments to verify that this is not an artefact caused by the FRET method or the assay. First, the mean amount of p-EGFR per endosome did increase at EGF concentrations higher than 10 ng/ml (*Figure 1—figure supplement 8*), indicating that the value measured is not artificially fixed, for example, by limited antigen accessibility. Second, a similar constant mean value of p-EGFR per endosomes was estimated with an independent method using the Tyr1068 antibody (*Figure 1—figure supplement 4*). Third, the narrow distribution of p-EGFR per endosome may simply reflect the sorting into endosomes of regular size. Whereas at 10 min the p-EGFR amount per endosome increased with the endosome area (*Figure 1—figure supplement 9A*, black curve), at 30 min (steady state, *Figure 1B*), the same mean amount was present in small and large endosomes alike (*Figure 1—figure supplement 9A*, red curve). Therefore, the amount of activated receptors per endosome is independent of endosome area. Finally, we verified that it is not a phenomenon peculiar to HeLa cells but also occurring in non-immortalized, non-cancer cell lines. Using the anti-phosphoTyr1068 antibody, we found that in primary mouse hepatocytes upon EGF stimulation the mean amount of p-EGFR per endosome saturated at ~20 min whereas the mean amount of EGF continued to grow (data not shown), indicating that the packaging of p-EGFR in endosomes is not peculiar to a signalling-aberrant cancerous cell line.

In which endocytic compartment was p-EGFR packaged so uniformly? Nearly 80% of p-EGFR colocalized with the early endosomal marker EEA1 throughout the time course (*Figure 2—figure supplement 1A*). Less than 10% of p-EGFR colocalized with APPL1 after 15 min showing that it passed this endosomal compartment (*Miaczynska et al., 2004*) (Miaczynska et al., submitted). Very little p-EGFR colocalized with LAMP-1, a marker of late endosomes and lysosomes (*Figure 2—figure supplement 1C*). One possibility is that the packages of p-EGFR may reflect incorporation into intra-luminal vesicles (ILV) of multi-vesicular bodies (MVB). This possibility was ruled out using a previously described differential detergent solubilisation method (*Malerød et al., 2007*). We could determine that a large fraction of EGFR was not accessible to antibodies upon digitonin permeabilization,

reflecting sequestration into ILVs (*Malerød et al., 2007*; *Piper and Katzmann, 2007*) (see Suppl. information and *Figure 2A,B*). In contrast, p-EGFR was always detectable suggesting that it was not within ILVs.

How do the kinetics of p-EGFR endosomal packaging compare with the kinetics of receptor dephosphorylation and ubiquitylation? After 10 min of EGF stimulation, the pool of p-EGFR in EEA1-positive endosomes (*Figure 2—figure supplement 1A*, red curve) decayed faster than total EGFR (*Figure 2—figure supplement 1A*, green curve; $\tau_{\text{decay EGFR}} = 56.03 \pm 5.72$, $\tau_{\text{decay p-EGFR}} = 37.08 \pm 4.19$), possibly due to de-phosphorylation or preferential removal of p-EGFR from early endosomes. The latter can be excluded since the fraction of p-EGFR in EEA1-positive endosomes remained almost constant throughout the time course (*Figure 2—figure supplement 1B*). EGFR ubiquitylation is required for its internalization into endosomes, and this is dependent on EGFR phosphorylation and the recruitment of the c-Cbl E3 ligase (*Sigismund et al., 2013*). To compare the levels of ubiquitylated EGFR (ub-EGFR) with those of p-EGFR within the endosomal system, we modified the FRET assay using an anti-ubiquitin antibody (*Figure 2—figure supplement 2A*). The kinetics of ub-EGFR were significantly different from those of p-EGFR. Whereas the levels of p-EGFR peaked at 15 min, ub-EGFR reached its maximum at 30 min after stimulation (*Figure 2C*, compare red with blue curves) and decreased more slowly than p-EGFR at later times, probably reflecting deubiquitylation prior to receptor sequestration into ILVs (*Piper and Katzmann, 2007*). These results are consistent with the fact that the appearance of p-EGFR precedes that of ub-EGFR (*Umebayashi et al., 2008*). Moreover, ub-EGFR had a similar distribution to that of EGFR (*Figure 2—figure supplement 2B*) but significantly wider than p-EGFR (*Figure 2—figure supplement 2C*), suggesting that the mechanisms responsible for stabilizing the mean levels of p-EGFR per endosome are not correlated with receptor ubiquitylation.

Our data suggest the existence of a saturable mechanism adjusting the amount of p-EGFR in each individual endosome. Such a constant mean amount may be due to the formation of small clusters within early endosomes. To test this possibility, we imaged the spatial distribution of p-EGFR in endosomes using the anti-EGFR phosphoTyr1068 antibody by super-resolution microscopy. Using direct Stochastic Optical Reconstruction Microscopy (dSTORM) (*Lampe et al., 2012*), we could indeed visualize clusters of p-EGFR (*Figure 2D*, left panel) that decreased in size between 10 and 30 min of EGF internalization, in agreement with the narrowing of p-EGFR distribution over time (*Figure 1D*). To determine the number of molecules in the clusters, we used two methods. First, we developed a new method to estimate the number of fluorescent molecules in light microscopy images by measuring the intensity fluctuations during photo-bleaching over time (for details see 'Materials and methods' and *Figure 2—figure supplement 3*). Based on the fluorescence signal from the anti-phosphoTyr1068 antibody and EGFR-GFP, we estimated an average of $102 \pm 38$ and $76 \pm 29$ (Mean $\pm$ SEM) molecules of EGFR and p-EGFR per endosome 30 min after EGF (10 ng/ml) internalization (*Figure 2—figure supplement 3*), corresponding to $707 \pm 265$ and $527 \pm 202$ molecules per $\mu m^3$ of endosomal volume (apparent, assessed by light microscopy), respectively. A hundred EGFR molecules would require ~12 clathrin-coated vesicles for delivery to endosomes (see 'Materials and methods'). We also estimated the total number of GFP-EGFR per cell and found values (29,000) well in agreement with previous estimates for HeLa cells (see 'Materials and methods' and *Figure 2—figure supplement 1A,B*). Second, based on the size of receptor from the PDB database (structure ID: 3NJP), we calculated that $83 \pm 25$ (Mean $\pm$ SEM, N = 1456) receptors could fit in the apparent area of p-EGFR visualized by dSTORM, a value which is remarkably in agreement with the fluorescence intensities estimates.

To further validate that the constant mean amount of p-EGFR per endosome corresponds to receptor clusters, we performed a focused RNAi screen on established components of the endosomal receptor sorting machinery (CHLCb, CHLCa, Hip1, Hip1R, Htt, Tom1, Tollip, Tom1L1, Tom1L2, Hrs, Snf8, Vps24). We found that only Hrs depletion resulted in a continuous accumulation of p-EGFR in endosomes with time (*Figure 2E*). At the same time, the p-EGFR intensity distribution widened similar to that of total EGFR (*Figure 2—figure supplement 4A,B*). The silencing of Hrs also caused an increase in the size of p-EGFR clusters within each endosome as revealed by dSTORM (*Figure 2D*, right panel). Interestingly, this is not due to the inhibition of ILV formation, as down-regulation of Snf8 and Vps24, members of the ESCRT-II and ESCRT-III complexes, respectively (*Piper and Katzmann, 2007*), reduced the sequestration of EGFR into the endosomal lumen (*Figure 2—figure supplement 4D*) but did not have significant effects on the amount of p-EGFR per endosome (*Figure 2E*, blue and green curves).

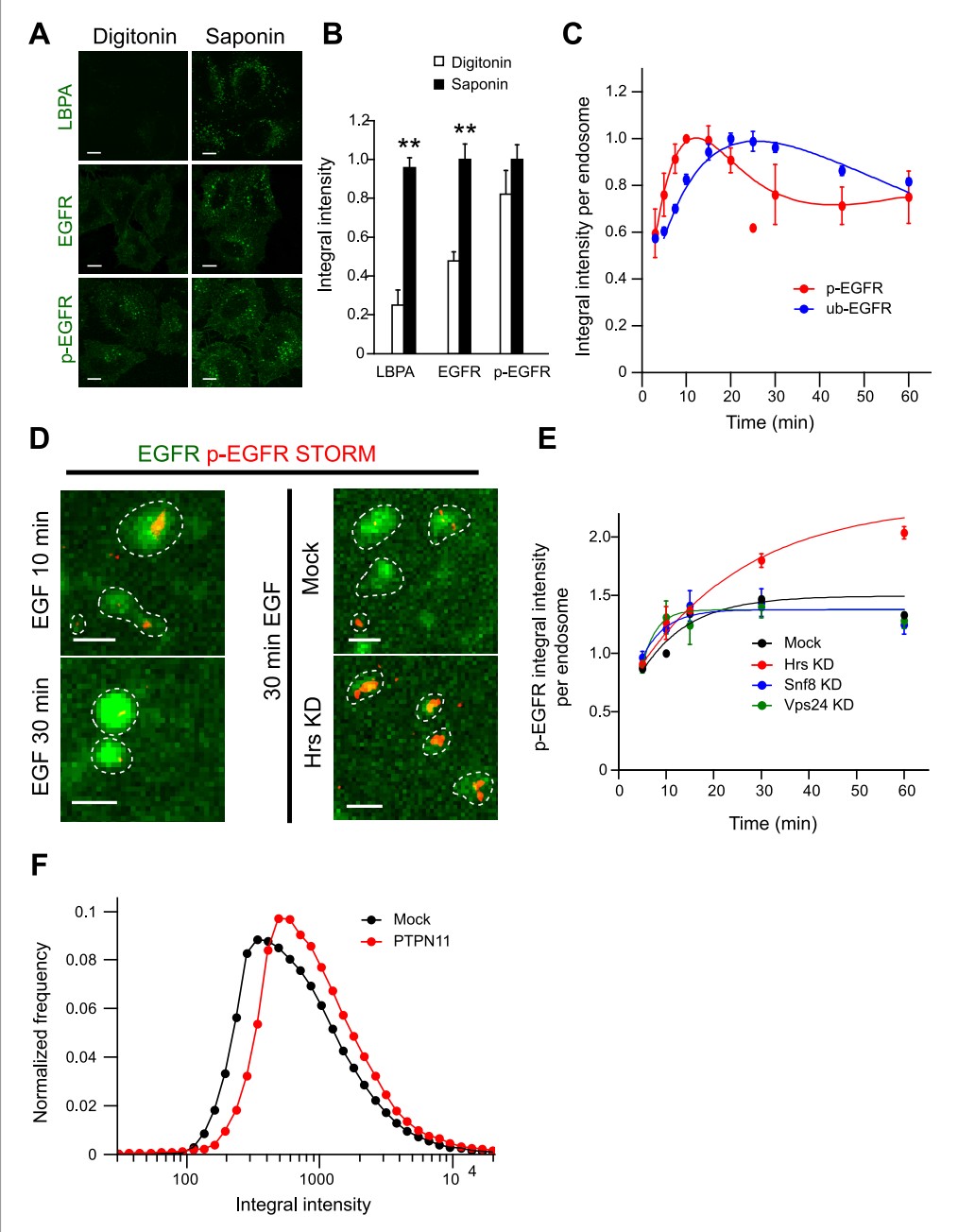

**Figure 2**. The constant mean amount of p-EGFR per endosome corresponds to receptor clusters that are regulated by Hrs and PTPN11. (**A**) Representative images of total EGFR and p-EGFR after staining with saponin or digitonin permeabilization methods. Immunofluorescence staining of LBPA is shown as a control marker for ILVs in MVBs. Scale bars, 10 μm. (**B**) Integral intensity of EGFR, p-EGFR, and LBPA (mean ± SEM) after permeabilization with digitonin or saponin. **p < 0.005 by a two-tailed *t*-test. Measurements were done in three independent experiments with a total of ∼150 cells per condition. (**C**) Time course of mean integral intensity per endosome for ub-EGFR (blue curve) upon EGF stimulation as in *Figure 1A*. p-EGFR is included for comparison. (**D**) Representative STORM images of p-EGFR (red) stained using a rabbit monoclonal anti-p-EGFR (Tyr 1068) antibody overlaid on top of a high magnification confocal image of EGFR (green). Left panels show clusters of p-EGFR upon stimulation with EGF for 10 or 30 min. Right panels show clusters of p-EGFR upon stimulation with EGF for 30 min in Hrs down-regulation or mock treatment. (**E**) Time course of the mean p-EGFR integral intensity per endosome in Hrs (red), Snf8 (blue), Vps24 (green), or mock-treated cells (black) (using three different siRNA oligonucleotides per gene). All curves were normalized by the intensity value at 10 min for the mock sample. Points show mean ± SEM from three different siRNAs per gene. Scale bar, 1 μm. (**F**) Integral intensity distribution of p-EGFR per endosome after down-regulation

*Figure 2. continued on next page*

*Figure 2. Continued*

for 72 hr of PTPN11 (red) or in mock treatment (black) after continuous stimulation with 10 ng/ml EGF for 30 min. Red points show the average distribution of three different siRNAs. Experimental points were fitted as in *Figure 1A*.

The following figure supplements are available for figure 2:

**Figure supplement 1**. The majority of the p-EGFR is located in EEA1-positive endosomes.

**Figure supplement 2**. ub-EGFR measurements by FRET microscopy.

**Figure supplement 3**. Quantification of number of EGFR and pEGFR molecules per endosome.

**Figure supplement 4**. Hrs, but not ESCRT-II or ESCRT-III components, increases the mean p-EGFR amount per endosome.

**Figure supplement 5**. Kinetics of Shc1 recruitment to endosomes.

**Figure supplement 6**. Pharmacological inhibition of EGFR kinase rapidly decreases the total p-EGFR in endosomes only at high but not low EGF concentrations.

**Figure supplement 7**. Pharmacological inhibition of EGFR kinase activity increases the mean p-EGFR amount per endosomes.

**Figure supplement 8**. Phosphatases can control the p-EGFR packaging in endosomes.

Thus, the constant mean amount of p-EGFR in endosomes likely corresponds to the receptor clusters observed by super-resolution microscopy. This raises the question of whether p-EGFR can be accessible to downstream signalling components. Therefore, we measured the recruitment of a direct downstream effector of p-EGFR, Shc1. The kinetics of Shc1 recruitment to endosomes precisely mimic the kinetics of p-EGFR (*Figure 2—figure supplement 5*) arguing that the p-EGFR clusters are signalling competent.

Upon internalization, EGF enters the early endosomal network and, similar to LDL (*Rink et al., 2005*), following endosome homotypic fusion and fission reactions, accumulates in few large endosomes prior to transfer to late endosomes. A mechanism must exist that prevents the continuous accretion of p-EGFR upon endosome fusion. A simple mechanism could be that the de-phosphorylation rate increases with the increase in p-EGFR per endosome. When two endosomes fuse, the resulting endosome should contain the sum of EGFR and p-EGFR of the original endosomes. However, given such de-phosphorylation rate dependency, the amount of p-EGFR would return to the level prior to fusion, thus stabilizing the mean amount of p-EGFR per endosome. A prediction of this hypothesis is that the kinase activity of EGFR in endosomes controls its own dephosphorylation. To test this, we inhibited the EGFR kinase activity pharmacologically with AG1478, lapatinib or gefitinib 10 min after EGF stimulation (to prevent alterations on receptor internalization) and determined the effects on the receptors already internalized and phosphorylated. We compared low with high concentrations of EGF, that is, under conditions of saturation of p-EGFR packaging in endosomes (*Figure 1—figure supplement 8*). At low EGF concentrations, when the packaging mechanism is not saturated, the total amount of p-EGFR was not significantly reduced by the inhibitors (*Figure 2—figure supplement 6* compare black and green curves). This behaviour argues that the packages of p-EGFR in endosomes are protected from the phosphatases. In addition, the inhibitors caused a continuous accumulation of p-EGFR in fewer and larger endosomes over time (*Figure 2—figure supplement 7*). In contrast, adding the inhibitor after stimulation with high concentrations of EGF caused a sharp reduction in the total amount of p-EGFR (*Figure 2—figure supplement 6* compare red and blue curves), as observed previously (*Kleiman et al., 2011*). This means that the kinase activity of EGFR is necessary to maintain the levels of p-EGFR in individual endosomes. These results support the idea that the dephosphorylation of p-EGFR in endosomes indeed depends on the EGFR activity within the endosomal packages.

Which phosphatases are responsible for controlling p-EGFR packaging in endosomes? To identify them, we performed a focused RNAi screen against 21 protein tyrosine phosphatases (PTP) expressed in HeLa cells (*Tarcic et al., 2009*). Hits were defined if silencing satisfied three conditions: (1) it increased the total amount of p-EGFR in endosomes and (2) increased the mean amount of p-EGFR per endosome, and (3) the phenotype was observed with at least two siRNAs per gene. Five phosphatases, PTP4A1, PTPN11, PTPN9, PTPN18, and PTPRK, increased the amount of p-EGFR in individual endosomes (*Figure 2F* and *Figure 2—figure supplement 8*). Interestingly, PTPN11 is an EGFR interactor (*Deribe et al., 2009*) whose activity is enhanced upon tyrosine phosphorylation (*Agazie and Hayman, 2003*), suggesting a molecular mechanism whereby p-EGFR could regulate its own de-phosphorylation in endosomes.

What are the consequences of such mechanism for signal transduction? To address these questions and generate testable predictions, we developed a mathematical model that describes the amount of total intracellular p-EGFR over time. Previously, excellent models have been developed that quantitatively describe EGFR endocytosis and signalling (*Felder et al., 1992*; *French et al., 1994*; *Kholodenko et al., 1999*; *Kholodenko, 2002*; *Resat et al., 2003*). However, although all these models described in detail the dynamics of ligand binding, dimer formation and endocytosis, recycling and degradation of the receptor, they did not consider the trafficking dynamics of the phosphorylated receptors with respect to the dynamics of the endosomal network because these data were not available. Our new experimental data brought two new concepts. First, dephosphorylation and degradation of p-EGFR occur sequentially but are uncoupled. Second, the amount of p-EGFR is controlled at the level of individual endosomes. These new concepts require further development of the existing EGFR mathematical models. Our model was formulated as a set of ordinary differential equations (ODE, see 'Materials and methods' and *Figure 3*) describing (1) the total amount of EGFR and p-EGFR at the plasma membrane as a function of ligand binding, (2) endocytosis of p-EGFR and its indirect effects on EGFR endocytosis, and (3) distribution of cargo between early endosomes at different stages of maturation (e.g., formation of MVB). For this, we considered the processes of receptor internalization, dephosphorylation, degradation, recycling, endosome fusion and fission. As in previous models (*French et al., 1994*; *Resat et al., 2003*), we described time course kinetics of total cellular p-EGFR, surface and endosomal EGFR and p-EGFR. Importantly, our model also describes the total number of p-EGFR-positive endosomes and mean amount of p-EGFR per endosome (see 'Materials and methods' for details). To account for the observed stabilization of the mean amount of p-EGFR per endosome over time (*Figure 1*), the dependency of p-EGFR dephosphorylation on EGFR kinase activity (*Figure 2—figure supplement 6,7*) and the fact that the mechanism is saturable (*Figure 1—figure supplement 8*), we included a sigmoidal dependency of the p-EGFR dephosphorylation rate on the amount of p-EGFR per endosome. The model was then fitted to the experimental data from the p-EGFR time course (*Figure 3A,B*). *Figure 3C* shows that this simple theoretical model can reproduce our observations of a constant mean amount of p-EGFR per endosome in a wide range of EGF concentrations when fitted to the experimental data. Importantly, a model without this non-linear dephosphorylation dependency could correctly describe the total amount of EGFR and p-EGFR in endosomes (*Figure 3—figure supplement 1A,B*) but did not agree with the measurements for the mean amount of p-EGFR per endosome (*Figure 3—figure supplement 1C*), thus supporting the sigmoidal dependency of the p-EGFR de-phosphorylation rate on the amount of p-EGFR per endosome (*Figure 3*). Previous models did not include this non-linear term because data on the distribution of p-EGFR in individual endosomes was not available.

An unexpected prediction of our model is that the total de-phosphorylation rate, and thus the total amount of p-EGFR, is dependent on the fusion/fission rate of the endosomes (*Figure 3D*). If so, could this have an effect on signal transduction? To test these hypotheses, we reduced early endosome homotypic fusion by lowering the intracellular concentration of established components of the endosome tethering and fusion machinery, EEA1, Rabenosyn5, Vps45 (*Christoforidis et al., 1999*; *Ohya et al., 2010*), Syntaxin-6 and Syntaxin-13 (*Brandhorst et al., 2006*) that play no direct role in signalling. These genes were down-regulated by RNAi in combinations and only partially (~50–70% depletion for each protein, *Figure 4A*) to achieve a significant inhibition of endosome fusion and yet prevent or reduce cell toxicity. This procedure caused a mild redistribution of EGFR to endosomes of smaller size (<0.5 $\mu m^2$ cross-section area, for details see 'Materials and methods' and *Figure 6—figure supplement 2*) (*Figure 4C,D*). Similar results were obtained upon depletion of a second combination of

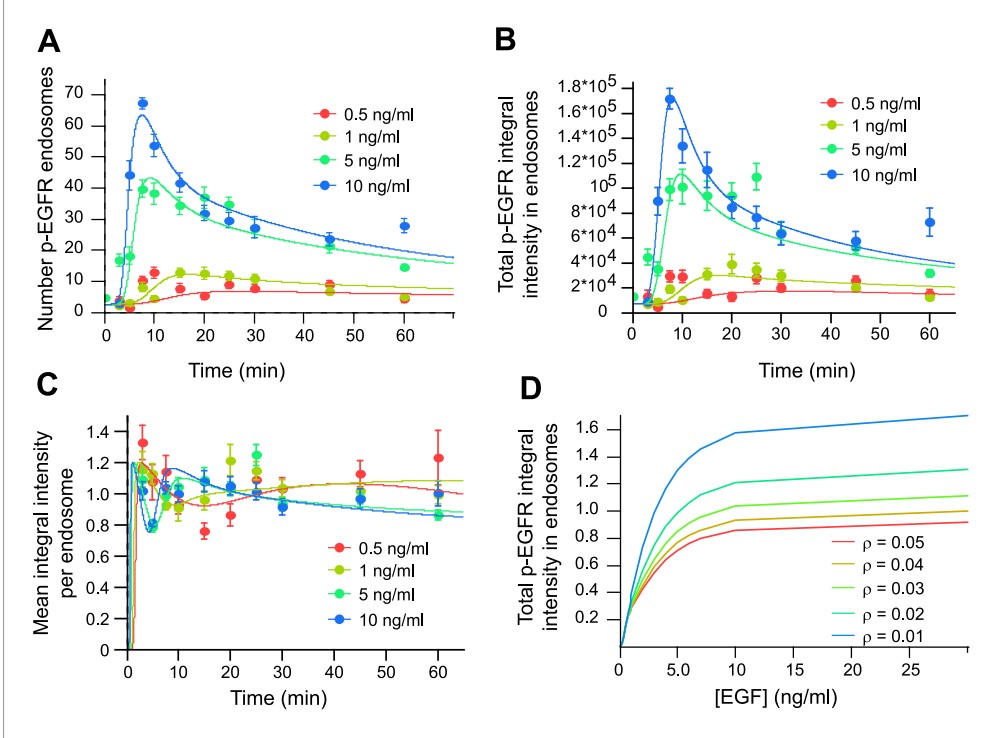

**Figure 3**. Mathematical model of p-EGFR predicts signalling amplitude and duration depends on early endosome fusion/fission rate. Parameters of the mathematical model were fitted to the experimentally measured number of p-EGFR endosomes, total integral intensity of p-EGFR, mean integral intensities of p-EGFR per endosome and total vesicular EGFR. The experimental data were obtained in a time course of EGF stimulation at four concentrations (0.5, 1.0, 5.0, and 10 ng/ml, colour coded as indicated). The fit results are presented on panels (**A**–**C**). The experimental data and model predictions are drawn as filled circles and solid curves, respectively. (**A**) Number of p-EGFR endosomes per 1000 $\mu m^2$ of cell area. (**B**) Total integral intensity of p-EGFR measured by FRET. The scaling factors that convert arbitrary numbers of the model to the experimental data were found by the least square procedure (see 'Materials and methods'). (**C**) Comparison of mean integral intensity of p-EGFR per endosome measured experimentally (filled circles) and mathematical model (solid curves) of the time course of p-EGFR upon EGF stimulation as in *Figure 1A*. The concentration of EGF is colour coded as presented. (**D**) Model predictions of the total amount of p-EGFR in endosomes as a function of EGF concentration and in the presence of different homotypic early endosome fusion rates (colour coded as indicated).

The following figure supplement is available for figure 3:

**Figure supplement 1**. A mathematical model without the non-linear phosphorylation dependency cannot describe the mean amount of p-EGFR per endosome.

genes (EEA1, Stx13, Stx6, not shown, see below). Note that these treatments generated a pattern of endosomes similar to that observed in different cell types and under different culture conditions (see below, *Figure 6*) and neither altered the surface levels of EGFR (*Figure 4—figure supplement 1*) nor its kinetics of uptake (*Figure 4B*) and exit from endosomes, that is, recycling and degradation (*Figure 4—figure supplement 2*). We also excluded potential effects on endosome acidification, because blocking it with bafilomycin did increase both p-EGFR and total EGFR (*Figure 4—figure supplement 3*). Remarkably, under our experimental conditions of mild down-regulation of the early endosomal fusion machinery the packaging of active receptors was unaffected as shown by both the time course and the steady-state mean (constant) amount of p-EGFR per endosome (*Figure 4E*; see above, *Figure 1B*). In contrast, the total number of endosomes with p-EGFR and their life-time augmented (*Figure 4F*), resulting in a net increase in the total amount and life-time of p-EGFR (*Figure 4G*). Notably, reduction of the endosome fusion rate in the mathematical model (~40%, in line with the depletion of tethering proteins, *Figure 4A*) is sufficient to reproduce fairly well the experimental increase in p-EGFR

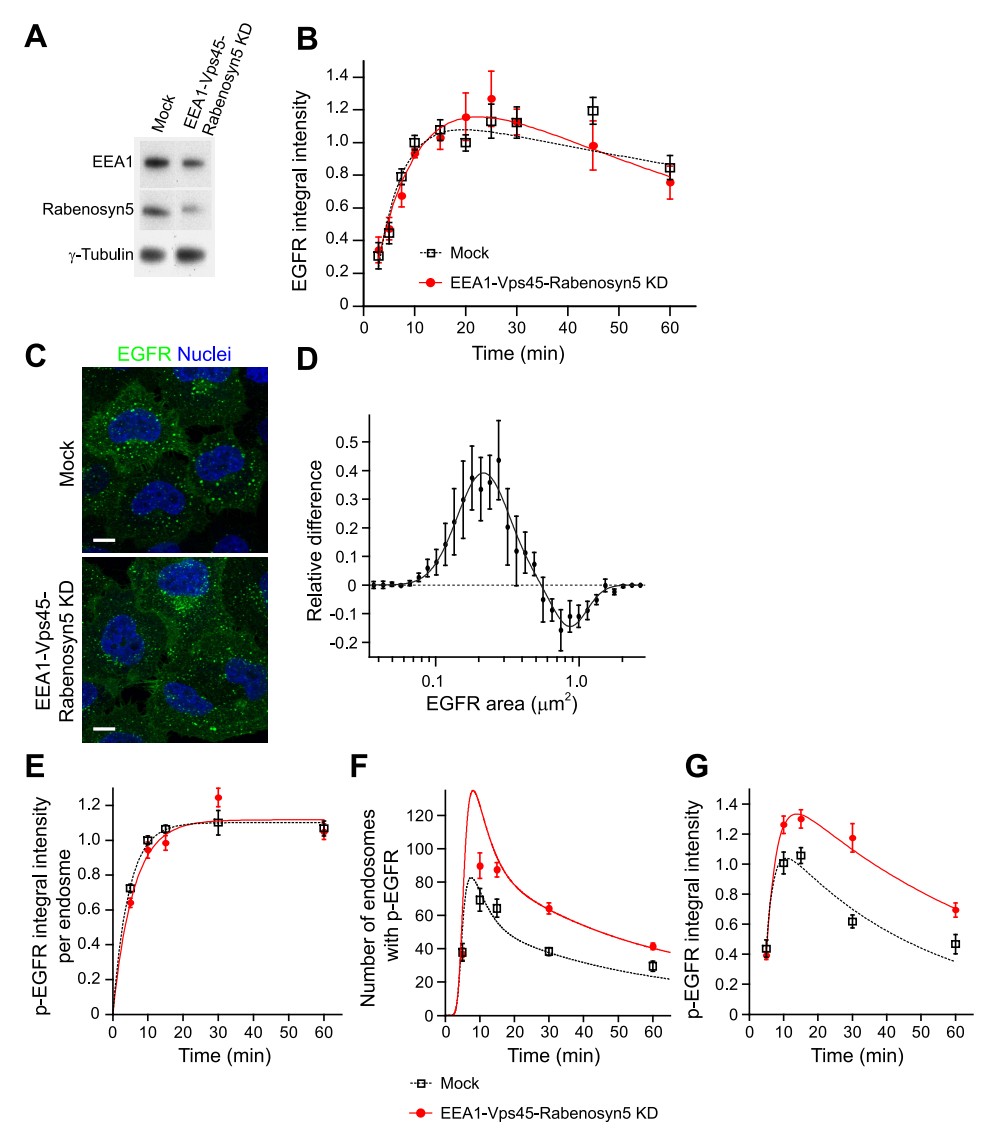

**Figure 4**. Increasing the number and life-time of p-EGFR endosomes results in prolonged EGFR activation. (**A**) Protein down-regulation of EEA1 and Rabenosyn5 72 hr after siRNA transfection. RT-PCR showed an 80% reduction in Vps45 mRNA levels (data not shown). (**B**) Time course of EGFR integral intensity in endosomes after partial protein depletion of EEA1, Rabenosyn5, and Vps45 (red curve) or mock treatment (black curve). Cells were given a 1-min pulse of 10 ng/ml EGF, washed and chased for the indicated time points before fixation. (**C**) Representative images of HeLa EGFR BAC cells after EEA1, Rabenosyn5, and Vps45 knock-down or treatment with transfection reagent only (mock). Scale bars, 10 μm. (**D**) Shift in the EGFR-endosome area distribution toward smaller endosomes after EEA1, Rabenosyn5, and Vps45 knock-down. The values of the histograms of endosome area distribution for the control and knock-down conditions were normalized and subtracted. The curve shows the relative increase (above zero) or reduction (below zero) in the number of endosomes for each area bin (in logarithmic scale) (for details see 'Materials and methods' and *Figure 6 - figure supplement 2*). Experimental points were fitted with two log-normal distributions. (**E–G**) Changes in p-EGFR endosomes in EEA1, Rabenosyn5, and Vps45 knock-down (red curve) or mock-treated (black curve) cells after continuous stimulation with 10 ng/ml EGF. Time courses of the mean integral intensity of p-EGFR per endosome (**E**), mean number of p-EGFR endosomes determined experimentally (squares) or predicted by the mathematical model (solid curves) for a 37% endosomes fusion rate (red curve) compared to control (black curve) (**F**), and total p-EGFR integral intensity in endosomes (**G**) measured as in *Figure 1*. Intensity curves were

*Figure 4. continued on next page*

*Figure 4. Continued*

normalized to the intensity value at 10 min for mock-treated cells. Experimental points show mean ± SEM. All measurements were done in three independent experiments with a total of ~150 cells per time point or condition. Time courses were fitted as in *Figure 1*.

The following figure supplements are available for figure 4:

**Figure supplement 1**. Knock-down of fusion machinery does not change EGFR distribution at the plasma membrane in HeLa cells.

**Figure supplement 2**. Knock-down of fusion machinery does not change EGFR degradation in HeLa cells.

**Figure supplement 3**. Blocking endosome acidification with Bafilomycin increases both total EGFR and p-EGFR, but not the mean amount of p-EGFR per endosome.

endosomes observed (*Figure 4F*). These results support the hypothesis that EGFR activation can be modulated by the endosomal system. Since p-EGFR de-phosphorylation precedes EGFR degradation (see above, *Figure 1A*, *Figure 2—figure supplement 2*) and EGFR degradation is unaffected (*Figure 4B*, *Figure 4—figure supplement 2*), we deduce that the effect on the life-time of p-EGFR caused by reduced endosomal fusion is primarily due to reduced de-phosphorylation.

Increased EGFR phosphorylation results in sustained Erk signalling (*Sasagawa et al., 2005*; *Nakakuki et al., 2010*) and this leads to the phosphorylation and stabilization of the immediate early gene product c-Fos (*Nakakuki et al., 2010*). We asked whether the redistribution of endosomal EGFR could be sufficient to induce sustained Erk activation and c-Fos phosphorylation. Indeed, upon EGF stimulation, both the amplitude and duration of Erk1/2 phosphorylation were increased in the depleted cells compared to control (*Figure 5A,B*). Consistently, c-Fos phosphorylation was also higher after 30 min of EGF stimulation (*Figure 5C,D*). The fact that the amount and life-time of total EGFR in endosomes remained unvaried in these experiments (*Figure 4B*) eliminates the trivial possibility that the observed changes are due to modulation of receptor degradation.

The experiments on HeLa cells and the theoretical analysis raise the question of whether modulation of early endosome homotypic fusion is a general mechanism to regulate signal amplitude and duration. If this were the case, we would predict that growth factors with different signalling outputs (amplitude and duration) differentially modulate the endosomal distribution (i.e., endosome number, size, and cargo content). To test this prediction, we examined different growth factors and cellular systems. First, we used primary mouse hepatoblasts where HGF promotes their proliferation (*Tanimizu et al., 2003*). In these cells, HGF but not EGF elicits a sustained Erk response (*Figure 6—figure supplement 1*). Indeed as predicted, stimulation of hepatoblasts with HGF caused a strong shift in the distribution of early endosomes toward smaller sizes (*Figure 6A*, red curve *Figure 6B*), whereas EGF had the opposite effect (*Figure 6A*, green curve, *Figure 6B*). Second, we turned to an in vitro model of reference for cell-fate decisions, PC12 cells. In PC12 cells, EGF stimulation leads to transient Erk phosphorylation and cell proliferation, whereas NGF leads to sustained Erk phosphorylation and cell differentiation (*Marshall, 1995*). Consistent with our results in primary mouse hepatoblasts, NGF stimulation in PC12 cells caused a significant shift in the distribution of early endosomes toward smaller sizes compared with EGF (*Figure 6C,D*). Moreover, NGF itself was distributed to a larger number of smaller endosomes in comparison with EGF (*Figure 6E,F*). Altogether, these data argue that the modulation of endosome fusion, reflected by the changes in endosome number and size, is a general property of growth factors. These data further suggest that signalling amplitude and duration can be regulated by changes in the fusion rate of endosomes (see *Table 1*).

Finally, we tested whether the differences in endosomal distribution can be, at least in part, causative of the different cell-fates triggered by EGF and NGF in PC12 cells. If so, we would predict that redistributing EGF to a larger number of small endosomes as seen in PC12 stimulated with NGF would be sufficient to switch signalling specificity and induce differentiation of PC12 cells. Therefore, we applied the same protocol of partial protein depletion previously used for HeLa cells (*Figure 4*) in PC12 cells and consistently observed a mild redistribution of EGF into smaller endosomes (*Figure 7—figure supplement 1A,C,D*). Also in this case, the partial depletion did not result in major

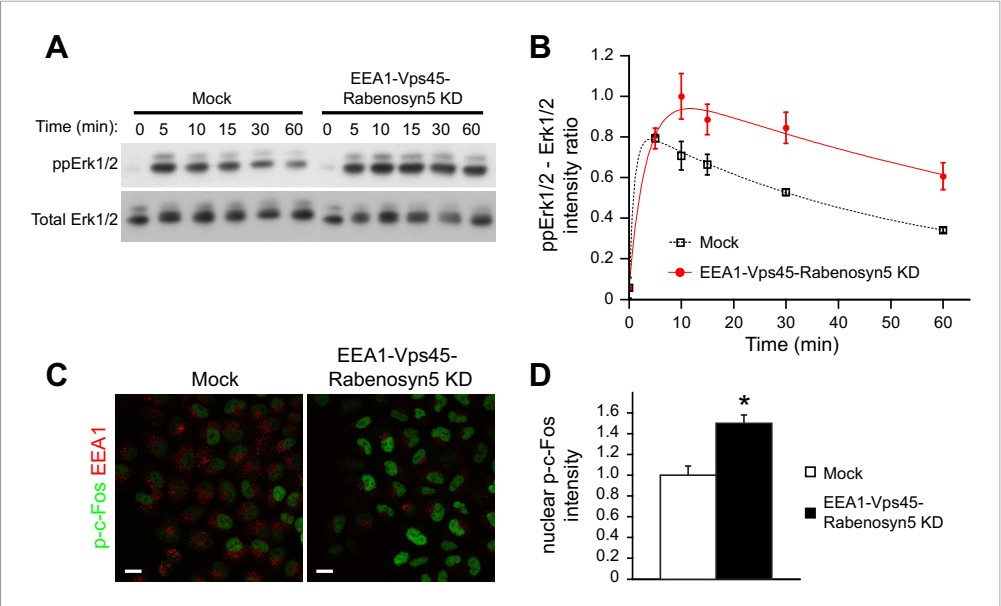

**Figure 5**. Redistribution of endosomal EGFR increases the amplitude and duration of MAPK signalling. (**A–B**) Time course of Erk1/2 phosphorylation after partial protein depletion of the three endosomal fusion components EEA1, Rabenosyn5, and Vps45 or mock treatment and continuous stimulation with 10 ng/ml EGF for the indicated times in HeLa EGFR BAC cells. (**A**) Representative phospho-Erk1/2 and Erk1/2 Western blots and (**B**) their quantification for EEA1, Rabenosyn5, and Vps45 knock-down (red curve) or mock-treated (black curve) samples. Points show mean ± SEM from three independent experiments. The time course was fitted as in *Figure 1*. (**C–D**) Nuclear c-Fos phosphorylation in EEA1, Rabenosyn5, and Vps45 knock-down or mock-treated cells as in (**A**) after 30 min of EGF stimulation. (**C**) Representative images of EEA1 and phospho-c-Fos immunostaining in EEA1, Rabenosyn5, and Vps45 knock-down or mock-treated cells. Scale bars, 20 μm. (**D**) Total intensity of nuclear phospho-c-Fos in EEA1, Rabenosyn5, and Vps45 knock-down or mock-treated cells. Bar graph shows mean ± SEM. Measurements were done in three independent experiments from a total of ~1000 cells per condition. *$p < 0.05$ by a 2-tailed t-test.

changes in EGF transport kinetics in PC12 cells (*Figure 7—figure supplement 1B*), but increased the phosphorylation of both Erk (*Figure 7—figure supplement 2A,B*) and c-Fos (*Figure 7—figure supplement 2C,D*) upon stimulation with EGF. Next, we stimulated PC12 cells with EGF or NGF for 24 hr and analysed for neurite formation and β-III tubulin expression as markers of differentiation (*Ohuchi et al., 2002*) and for EdU incorporation as a measure of proliferation (*Figure 7A*). Stimulation with NGF increased the number of cells with neurites (*Figure 7B*, quantification in *Figure 7C*) and positive for β-III tubulin (*Figure 7B*, quantification in *Figure 7D*), and reduced cell proliferation (*Figure 7A*, quantification in *Figure 7E*), the opposite of the stimulation with EGF. Remarkably, upon redistribution of endosomes, EGF increased process formation (*Figure 7B*, quantification in *Figure 7C*), β-III tubulin expression (*Figure 7B*, quantification in *Figure 7D*), and reduced cell proliferation (*Figure 7A*, quantification in *Figure 7E*). The type of response was therefore similar to that of NGF, although the efficacy was lower. Nevertheless, these results show that a mild reduction of homotypic early endosome fusion was sufficient to modify cell fate and induce neuronal differentiation of PC12 cells.

## Discussion

Genomic studies have revealed that signalling pathways exert a profound effect on the endosomal system (*Pelkmans et al., 2005*; *Stasyk et al., 2007*; *Collinet et al., 2010*). Parameters such as number of endosomes and size are tightly controlled in the case of EGF endocytosis (*Collinet et al., 2010*). Our results provide a rationale for such modulation and a novel framework for interpreting and predicting the signalling response of phosphorylated RTKs. In homogeneous assays (e.g., by Western blot), the total levels of active RTKs can be observed to rapidly decay with time in most signalling systems (*Dunn and Hubbard, 1984*; *Burke et al., 2001*; *Sousa et al., 2012*). These methods, however, measure the average steady state of an entire cell population and lack the spatial

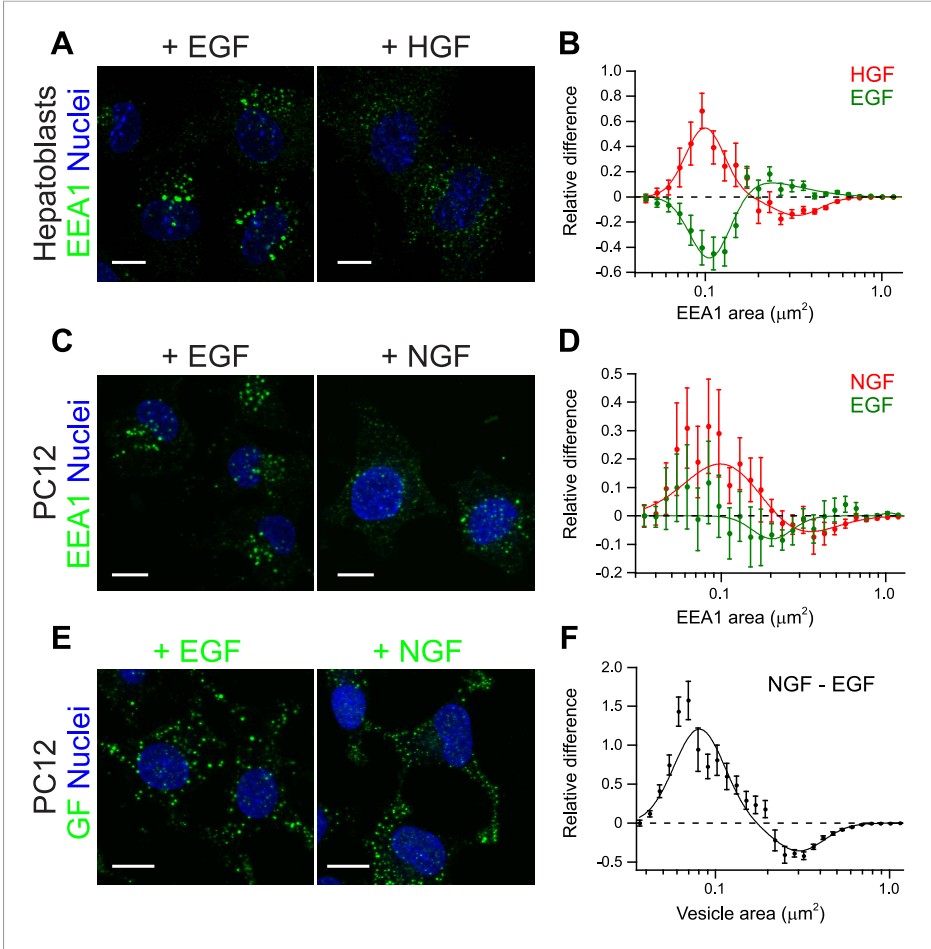

**Figure 6**. Growth factors differentially shift the distribution of the number and size of endosomes. (**A**) Representative images of primary mouse hepatoblasts after stimulation with 10 ng/ml EGF or HGF for 30 min (**B**) Shift in the EEA1-positive endosome area distribution after stimulation with HGF (red curve) or EGF (green curve). The values of the histograms of endosome area distribution for growth factor stimulated and non-stimulated cells were normalized and subtracted. The curve shows the relative increase (above zero) or reduction (below zero) in the number of endosomes for each area bin (in logarithmic scale). HGF stimulation increased while EGF decreased the proportion of endosomes smaller than $0.2\ \mu m^2$. (**C–F**) PC12 cells after stimulation for 30 min with 100 ng/ml EGF or 50 ng/ml NGF. (**C**) Representative images of EEA1-positive endosomes. (**D**) Shift in the EEA1-positive endosome area distribution after stimulation with NGF (red curve) or EGF (green curve) measured as in (**B**). NGF stimulation increased while EGF slightly decreased the proportion of endosomes smaller than $0.2\ \mu m^2$. (**E**) Representative images of EGF or NGF. (**F**) Differences in the area distribution of endosomal NGF and EGF measured as in (**B**). NGF is enriched in endosomes smaller than $0.2\ \mu m^2$ relative to EGF. For all graphs points show the mean ± SEM of experimental distributions. Measurements were done in three independent experiments with n ~150 cells per condition. In all graphs, experimental points were fitted with two log-normal distributions. Image scale bars, 10 μm.

The following figure supplements are available for figure 6:

**Figure supplement 1**. HGF triggers sustained Erk1/2 activation in primary mouse hepatoblasts.

**Figure supplement 2**. Quantification of the difference between two area distributions.

information. Here, we employed quantitative high- and super-resolution microscopy to resolve details of this process with sub-cellular resolution and high sensitivity. We discovered that the mean amount of p-EGFR per endosome was fairly constant over time and p-EGFR was found in small clusters in early endosomes.

**Table 1**. Changes in endosome number and area

| Cell type | Endosome marker or cargo | Growth factor | Endosome number* | Endosome area (µm²) | Increase in number of smaller vesicles |
|---|---|---|---|---|---|
| HeLa# | EGFR | EGF | 22 ± 9 | 0.518 ± 0.023 (control = 0.629 ± 0.029) | 9.53% ± 0.014 (<0.4 µm²) |
| E14.5 hepatoblast | EEA1 | EGF | −6 ± 18 | 0.286 ± 0.02 (control = 0.294 ± 0.02) | −0.91% ± 0.003 (<0.3 µm²) |
| E14.5 hepatoblast | EEA1 | HGF | 18 ± 12 | 0.276 ± 0.02 (control = 0.294 ± 0.02) | 2.05% ± 0.002 (<0.3 µm²) |
| PC12 | EEA1 | EGF | 3 ± 10 | 0.471 ± 0.05 (control = 0.461 ± 0.05) | −1.03% ± 0.01 (<0.3 µm²) |
| PC12 | EEA1 | NGF | 23 ± 16 | 0.454 ± 0.05 (control = 0.461 ± 0.05) | 2.2% ± 0.01 (<0.3 µm²) |
| PC12 | EGF | EGF | 316 ± 46 | 0.276 ± 0.003 | – |
| PC12 | NGF | NGF | 341 ± 5 | 0.245 ± 0.007 | 5.15% ± 0.01 (<0.3 µm², difference from EGF-endosomes) |

*Endosome number is expressed as the difference from the control or non-stimulated cells. The value shows the number of endosomes per 1000 µm² of area covered by cells.
#HeLa cells after knock-down of EEA1, Rabenosyn5, and Vps45. All values show mean ± SEM.

The endosomal network is shaped by the balance of endosome fusion and fission (*Foret et al., 2012*) and this balance is also necessary for the formation of the endosomal clusters of p-EGFR. Modulation of the endosomal fusion/fission machinery manifests itself as a change in the size of endosomes (*Sigismund et al., 2008*) (*Figure 6*). Shifting the balance toward smaller endosomes through inhibition of fusion increased the number and reduced the size of endosomes, consequently expanding the number and life-time of p-EGFR clusters. Although the inhibition of endosome fusion was very mild, we cannot exclude the possibility that it may alter the recruitment and/or activity of signalling components by yet unknown mechanisms. On the other hand, for the interpretation of phenotypes upon perturbations on signalling, it is also important to consider the impact they have on the endosomal network (*Collinet et al., 2010*).

By analogy with synaptic transmission (*Edwards, 2007*), the packages of p-EGFR in early endosomes could be considered as *quanta* of signalling molecules. The concept of phosphorylated RTK *quanta* is reminiscent of analogue-to-digital communication systems, where a continuous variable (e.g., extracellular growth factor concentration) is transformed into a sequence of binary levels (e.g., phosphorylated RTK *quanta* in endosomes). An analogue-to-digital switch was described for Ras nanoclusters at the plasma membrane (*Tian et al., 2007*). In the case of endosomal digital signalling, our mathematical model predicts that it could serve two functions. First, it provides a mechanism to regulate signal amplitude and duration following RTK internalization. As a consequence, the total de-phosphorylation rate becomes dependent on the fusion/fission rate of the endosomes. This is interesting in view of the specific modulation of the endosome fusion/fission rates by growth factors (*Figure 6*, see below). Second, it acts as a noise dampening system (*Ladbury and Arold, 2012*), suppressing the noise due to, for example, fluctuations of EGF in the extracellular medium, expression levels of EGFR on the cell surface, etc. An increase in the amount of p-EGFR would result in faster de-phosphorylation rates. In contrast, low concentrations of EGF or EGFR would result in low de-phosphorylation rates. The middle point between the two extremes is the hallmark of signalling resilience. In addition, such a digital system may facilitate the integration of signalling information from different RTKs into a single, correct cell-fate decision. Our results highlight the importance of measuring the spatio-temporal distribution of signalling molecules using quantitative image analysis approaches to gain a deeper understanding of signal transduction regulation.

What is the molecular machinery responsible for the formation of the clusters and how is the number of p-EGFR molecules regulated? Clearly, the clustering mechanism is saturable (*Figure 2A,B*), as very high concentrations of EGF above some threshold suppress the correct endosomal packaging in addition to changes in the entry routes and signal output (*Sigismund et al., 2008*). We found that both Hrs and a few phosphatases, notably PTPN11 (SHP2), specifically regulate the amount of receptors within the p-EGFR clusters and their size. Hrs is known to interact with EGFR and regulate its

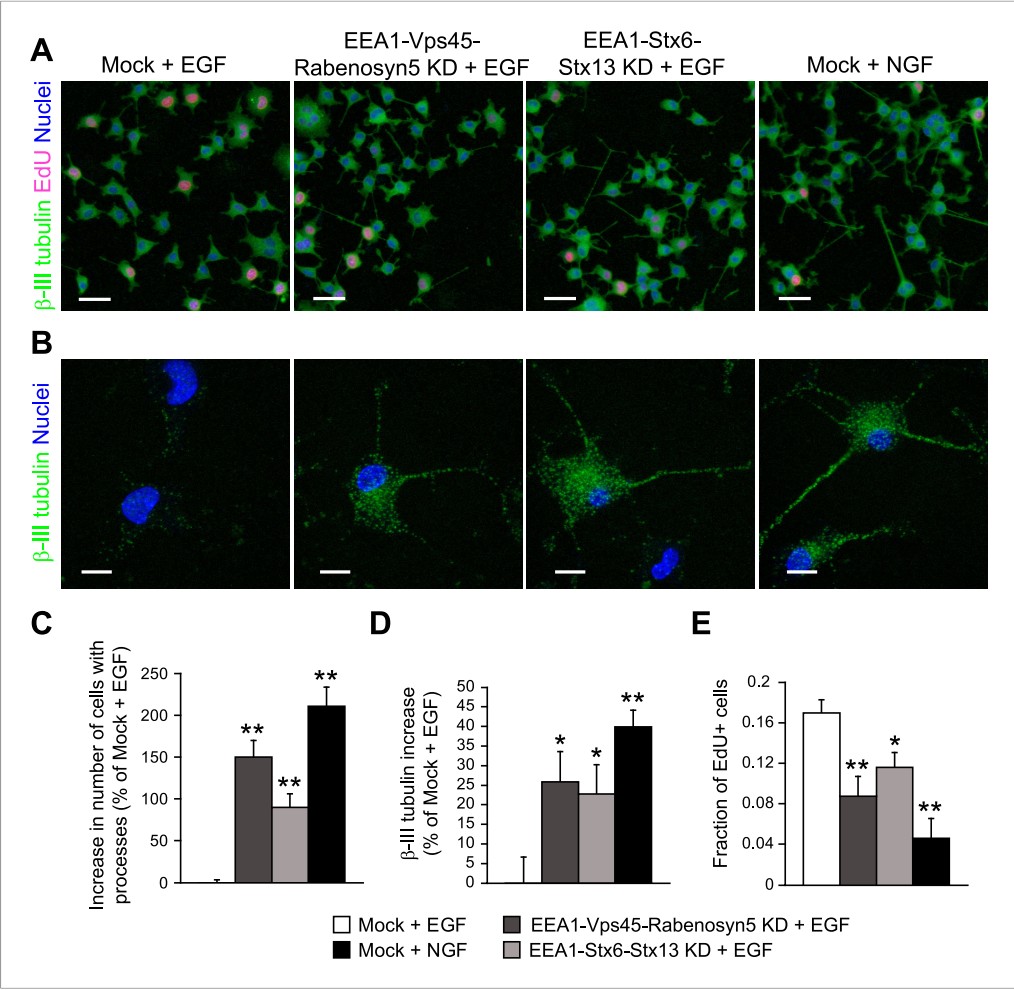

**Figure 7**. Redistribution of endosomal EGF is sufficient to trigger neuronal differentiation in PC12 cells. (**A–B**) Representative images of PC12 cells after partial protein depletion of either EEA1, Rabenosyn5, and Vps45 or EEA1, Syntaxin-6, and Syntaxin-13, or mock treatment and stimulation with 100 ng/ml EGF or 50 ng/ml NGF for 24 hr. Scale bars, 50 μm. (**B**) A high-resolution image of single cells to highlight the changes in β-III tubulin expression and neurite formation. β-III tubulin is shown in green, nuclei are shown in blue, and EdU-positive nuclei are shown in pink. Scale bars, 10 μm. Note that in *Figure 6C,E*, the short incubation times did not permit neurite outgrowth. (**C**) Increase in the number of cells with β-III tubulin-positive processes longer than 1 μm compared to mock-treated cells after EGF stimulation. (**D**) Increase in β-III tubulin expression measured by the total intensity of the cytoplasmic β-III tubulin immunostaining. The total intensity per image was normalized by the image area covered by cells. (**E**) Number of proliferating cells measured by EdU incorporation. The number of EdU-positive nuclei was divided by the total number of nuclei. In all cases, data show mean ± SEM. For each parameter, pair-wise comparisons were done against EGF-stimulated mock-treated cells. *$p < 0.05$, **$p < 0.005$ by Fisher's LSD test. All measurements were done in three independent experiments with a total of ~15000 cells per condition.

The following figure supplements are available for figure 7:

**Figure supplement 1**. Knock-down of fusion machinery redistributes endosomal EGF in PC12 cells.

**Figure supplement 2**. Redistribution of endosomal EGF is sufficient to increase MAPK activation in PC12 cells.

degradation together with other components of the ESCRT machinery (*Umebayashi et al., 2008*). However, the effect of Hrs on the size of the p-EGFR clusters appears to be independent of the formation of ILVs, as suggested by the fact that Snf8 and Vps24 down-regulation does not produce the same effect.

Our mathematical model revealed that a correlation between p-EGFR dephosphorylation rate and p-EGFR amount per endosome can explain the mean constant size of p-EGFR *quanta*. We can envisage various non-exclusive mechanisms that can account for this correlation. One possible mechanism is a scaffold with a characteristic size that binds to p-EGFR and protects it from phosphatases. This hypothesis correlates higher total EGFR kinase activity to higher p-EGFR dephosphorylation, but only indirectly. Increasing the concentration of EGF in the medium would lead to a higher rate of delivery of p-EGFR to endosomes through vesicles which have no scaffold. If scaffold formation were rate limiting, the increased flux of p-EGFR into endosomes would reduce the fraction of protected p-EGFR thus exposing it to dephosphorylation. A caveat of this model is that, as the fusion of endosomes proceeds over time, multiple *quanta* would be expected to be brought together, increasing the mean amount of p-EGFR per endosome. This expectation is in contradiction with our experimental data (*Figure 1B,D*). With this model, additional factors must thus be taken into account to explain why multiple *quanta* cannot co-exist on the same endosomes.

The finding that Hrs knock-down increases the levels of p-EGFR suggests a different scaffold-based model. Instead of acting as a p-EGFR protective scaffold (or part of a scaffold), Hrs could exert the opposite function and stabilize the unphosphorylated EGFR, preventing its re-phosphorylation (*Kleiman et al., 2011*). Since the activity of Hrs is negatively regulated by p-EGFR (*Row et al., 2005*; *Bache et al., 2002*), this model is compatible with the data showing loss of *quanta* and increase in endosomal p-EGFR levels upon Hrs knock-down (*Figure 2D,E*). However, this hypothesis alone can neither explain the formation of *quanta* nor the finding that blocking p-EGFR kinase activity does not change the total levels of p-EGFR over time (*Figure 2—Figure supplement 6*).

Another mechanism is based on Turing Instability (*Turing, 1952*) (a reaction-diffusion mechanism). This mechanism is perhaps less intuitive but widely spread in biological processes, such as symmetry breaking and pattern formation in morphogenesis (*Kondo and Miura, 2010*). It is based on the observation that p-EGFR recruits and phosphorylates PTPN11 (SHP2) in a phosphor-tyrosine dependent manner (*Deribe et al., 2009*), thus enhancing its phosphatase activity (*Agazie and Hayman, 2003*). Briefly, p-EGFR would recruit and activate the phosphatase SHP2, forming a negative feedback loop. The phosphatase would diffuse on the surface of endosomes, dephosphorylating p-EGFR molecules before being itself inactivated in the absence of further interactions with p-EGFR. Such reaction-diffusion mechanism within a specific parameter range is known to form spatially restricted clusters of active molecular species (Turing Instability) (*Turing, 1952*), in this particular case *quanta* of p-EGFR on endosomes. A transient increase in p-EGFR after an endosome fusion event would increase the recruitment and/or activity of SHP2, re-establishing the p-EGFR *quanta* through dephosphorylation. If the characteristic length of Turing Instability is larger than endosome size, then multiple *quanta* cannot co-exist within a single endosome. The Turing Instability hypothesis explains the observed increase in p-EGFR *quanta* size after EGFR kinase inhibition, keeping the total p-EGFR levels unchanged (*Figure 2—figure supplement 6,7*), as well as the increase in total endosomal p-EGFR upon inhibition of endosome fusion (*Figure 4G*). However, it does not explain the effect of Hrs knock-down. A combination of the Turing instability and Hrs-mediated (negative) scaffold mechanisms is more consistent with our observations.

The regulation of endosomal packing reported in our study is likely not restricted to EGFR alone but is a general property, as different growth factors affect the endosomal network according to their specific signal output and cellular context (*Figure 6*). Hrs and SHP2 are also recruited by other RTKs (*Agazie and Hayman, 2003*; *Row et al., 2005*). The relative affinity of SHP2 to different receptors could lead to larger or smaller quanta, thus tuning the specificity of the signalling response. RTK quanta with different sizes could also result from differential phosphorylation of Hrs by RTKs (*Row et al., 2005*), given that the relative amount of Hrs on endosomes depends on its phosphorylation state (*Urbé et al., 2000*).

By which mechanisms can RTKs regulate the endosomal network? It has been shown that RTKs can modulate the activity of the transport machinery. For example, activation of p38 MAP kinase causes phosphorylation of the Rab5 effectors EEA1 and Rabenosyn-5, enhancing their recruitment to endosomes and consequently stimulating early endosome fusion (*Macé et al., 2005*; *Cavalli et al., 2001*). RTK stimulation also modulates the nucleotide cycle of Rab5 via activation of the Rab5 GEF RIN1 (*Tall et al., 2001*) or inactivation of the Rab5 GAP RN-tre (*Lanzetti et al., 2000*). Therefore, we predict that in general RTK ligands that stimulate the endosomal fusion machinery (such as EGF) will have a short phosphorylation half-life, whereas ligands that change the fusion/fission balance in favour

of smaller endosomes (such as NGF) will have a long phosphorylation half-life. The combined effects of *quanta* size regulation through Hrs and SHP2 and modulation of fusion/fission will give a specific signalling amplitude and duration in different cell types stimulated with different ligands. We propose that the shape of distribution of the endosomal network can serve as a predictive parameter of the signalling status of the cell.

Our results support the concept of endosomes as signalling platforms (*Di Guglielmo et al., 1994*), a view recently shared for the β2-adrenoceptor (*Irannejad et al., 2013*) but opposed by other studies (*Brankatschk et al., 2012*; *Sousa et al., 2012*). This apparent contradiction can be explained by the fact that, under normal conditions of endocytosis, only the small fraction of p-RTK in endosomes are protected from inactivation and degradation, and can thus contribute to signal propagation. A feature of the p-EGFR clusters is that, with the increase in the local concentration, the stability of the active EGFR dimer (*Chung et al., 2010*) and signalling properties (*Verveer et al., 2000*) would also be increased. By blocking endocytosis, the levels of active receptors are artificially increased at the cell surface (*Sousa et al., 2012*), bypassing the normal requirement for endosomal regulation.

Our observations raise many more questions concerning the molecular mechanisms of *quanta* formation and their impact on cell fate decision. Clearly, the variety of models on *quanta* formation requires future experimental tests to determine the correct mechanism and reveal its molecular details. In addition, it will be important to validate our observations in an in vivo animal model to demonstrate that the dynamics of the endosomal network reflect the signalling activity by RTK under physiological conditions.

## Materials and methods

### p-EGFR FRET microscopy assay

To reduce the consequences of EGFR overexpression, we used HeLa Kyoto cells transfected with a bacterial artificial chromosome (BAC) transgene stably expressing EGFR-GFP under its endogenous promoter (*Poser et al., 2008*). Cells were incubated for different times in serum-free medium with 10 ng/ml EGF (Invitrogen, California, USA) or for 30 min with 0.05, 0.1, 0.25, 0.5, 1, 2.5, 5, 7.5, or 10 ng/ml EGF. Cells were then fixed and processed for immunofluorescence as previously described (*Collinet et al., 2010*) using a mouse monoclonal anti-phospho-tyrosine 4G10 antibody (Millipore, California, USA) directly labelled with AlexaFluor 555 (Molecular Probes, Invitrogen). For colocalization measurements, samples were also incubated with rabbit polyclonal anti-EEA1 (*Rink et al., 2005*) or mouse monoclonal anti-LAMP-1 (BD Biosciencies, California, USA) antibodies. Images were acquired using a laser-scanning confocal microscope (Duoscan, Zeiss) with a 63×/1.4 oil objective. Multicolour images were acquired in three sequential scans: GFP fluorescence and AlexaFluor 555 fluorescence were detected simultaneously with two different detectors using 488 and 561 nm laser light and a 505/530 band-pass filter or a 593 nm long-pass spectral range in a META detector (Zeiss); FRET signal was detected with 458 nm excitation and a 593 nm long-pass spectral range in a META detector (Zeiss). 10 images per time point were collected, and each image was the maximum projection of four confocal sections of ~1 μm thickness with 0.5 μm step. For the comparison between live and fixed cells, images were acquired with an automated spinning-disk confocal microscope (OPERA, Evotec Technologies-PerkinElmer) with a 40×/0.9 NA water immersion objective. EGFR-GFP was excited with a 488 nm laser and detected with a 520/35 nm filter. DAPI for nuclei identification was excited in a separate exposure with a 405 nm laser and detected a 450/50 nm filter. Eighty images per condition were acquired. Every image contained on average 20 cells.

Image analysis was performed using custom designed image analysis software (MotionTracking) as previously described (*Rink et al., 2005*; *Collinet et al., 2010*). The '*integral intensity*' corresponds to the integral of fluorescent marker intensity per endosome. The '*total integral intensity*' is defined as the sum of integral intensities of all endosomes in an image normalized by the area covered by the cells. The '*endosome cross-sectional area*' was measured as the apparent fluorescent area (in $\mu m^2$) of an endosome (above the half-maximum value of fluorescence intensity of each structure). Since MotionTracking approximates real image intensity by a sum of analytical functions (*Rink et al., 2005*), the resulting area and intensity have no pixel granularity.

## High-resolution microscopy FRET-based assay

p-EGFR or ub-EGFR was first identified on the basis of triple colocalization between objects detected by the EGFR (488 nm laser excitation and 505/530 nm bandpath emission filter), anti-p-Tyr antibody p-Tyr-ab for p-EGFR or anti-mono and polyubiquitynilated conjugates (FK2) (Enzo Biosciences, New York, USA) (561 nm laser excitation and a 593 nm long-pass filter), and FRET (458 nm laser excitation and a 593 nm long-pass filter) channels (*Figure 1—figure supplement 1A*). Colocalization was scored by cross-sectional overlap >30%. The FRET signal was corrected for spectral bleed-through (SBT). Two major processes contribute to the SBT: (1) the GFP fluorescence bleed-through in the FRET channel and (2) direct excitation of AlexaFluor 555 by the 458 nm laser. We performed control experiments to estimate SBT for subsequent correction. To estimate GFP fluorescence bleed-through, we imaged EGFR-GFP BAC HeLa cells in the FRET channel (excitation 458 nm) without p-Tyr-ab staining. The signal in the FRET channel was below our detection limit and, therefore, we omitted correction in the subsequent analysis. The SBT by direct excitation of AlexaFluor 555 was estimated by quantification of FRET vesicles that colocalized with p-Tyr-ab or ub-ab, but not with EGFR (bleed-through control). Following the approach of *Gordon et al., (1998)*, the correction in this case will be,

$$F = I - k \cdot T, \tag{1}$$

where $F$ is the corrected intensity in the FRET channel, $I$ is the raw intensity in the FRET channel, $T$ is the intensity in the p-Tyr-ab channel, $k = \frac{<I_{control}>}{<T_{control}>}$ is the bleed-through coefficient (ratio of means) calculated from control vesicles. Unfortunately, this correction method provided a good estimation of the average FRET signal, but when applied to individual endosomes it gave negative intensities for a substantial (30–40%) number of cases, thus precluding the estimation of mean intensity per endosome. In order to identify the source of negative intensities, we calculated the distribution of ratios of intensities in the FRET channel to the intensities in the p-Tyr-ab channel per endosome (*Figure 1—figure supplement 1B*). This distribution is broad and one can conclude that correction by *Equation 1* will inevitably produce negative values in some cases. We fitted the distribution by the sum of three Gaussian components (*Figure 1—figure supplement 1B*, red, green, and blue dashed lines). By using control cells that did not express EGFR-GFP, we tested that the first two components correspond to SBT (direct excitation of AlexaFluor 555 by the 458 nm laser). Next, we developed a probabilistic model to find the expected FRET signal, given the p-Tyr-ab signal and the constants ($\mu$, $\sigma$) of Gaussian distribution of the intensities ratios. The distribution of ratios is $P(m)dm = \frac{1}{\sqrt{2\pi}\sigma}e^{-\frac{(m-\mu)^2}{2\sigma^2}}dm$. We denoted $m = \frac{I-F}{T}$ the ratio of bleed-through signal in the FRET channel to the signal in the p-Tyr-ab channel for individual endosomes. After this substitution, the probability to obtain a FRET signal $F$ is $P(F)dF = P(m(F))\frac{dm}{dF}dF = \frac{1}{\sqrt{2\pi}\sigma T}\exp\left(\frac{-(F-(I-\mu T))^2}{2\sigma^2 T^2}\right)dF$. Since the bleed-through cannot be higher than the measured signal $I$, we can calculate the expectation of the FRET signal as: $\langle F \rangle = \frac{\int_0^I F \cdot P(F)dF}{\int_0^I P(F)dF}$. After substitution and integration, we get:

$$\langle F \rangle = I - \mu T + \sqrt{\frac{2}{\pi}}\sigma T \frac{\left(1 - e^{-\frac{1}{2}\left(\frac{\mu}{\sigma}\right)^2}\right) - \text{sgn}(I-\mu T) \cdot \left(1 - e^{-\frac{1}{2}\left(\frac{I-\mu T}{\sigma T}\right)^2}\right)}{\text{erf}\left(\frac{\mu}{\sqrt{2}\sigma}\right) + \text{erf}\left(\frac{I-\mu T}{\sqrt{2}\sigma T}\right)}. \tag{2}$$

One can see that (a) if $I - \mu T > 0$ (i.e., SBT is small relative to the true FRET signal), then the last term in the formula is very small and $\langle F \rangle \approx I - \mu T$ in agreement with Gordons' formula; (b) even if $I - \mu T < 0$ (i.e., SBT is large relative to the true FRET signal or the FRET signal is absent), the *Equation 2* always gives small, but positive values. As such, *Equation 2* provides a good estimation of the expected FRET intensity given the measured intensities in the FRET and p-Tyr-ab channels.

Next, we developed this approach further by taking into account that the real bleed-through distribution is the sum of two Gaussians with mean values $\mu_1$, $\mu_2$, standard deviations $\sigma_1$, $\sigma_2$ and their contribution in the total distribution $a_1$, $a_2$. Following the same approach as above we get:

$$\langle F \rangle = I - (a_1\mu_1 + a_2\mu_2)T + \sqrt{\frac{2}{\pi}(a_1^2\sigma_1^2 + a_2^2\sigma_2^2)T} \frac{(1 - e^{-M}) - \text{sgn}(N) \cdot (1 - e^{-N^2})}{erf(\sqrt{M}) + erf(N)}, \tag{3}$$

where $M = \left(\frac{\mu_1}{\sqrt{2}\sigma_1}\right)^2 + \left(\frac{\mu_2}{\sqrt{2}\sigma_2}\right)^2$ and $N = \frac{I - (a_1\mu_1 + a_2\mu_2)T}{\sqrt{2(a_1^2\sigma_1^2 + a_2^2\sigma_2^2)T}}$.

The example of FRET correction by *Equation 3* is presented on *Figure 1C*.

To validate the FRET measurements, cells were treated with EGF, stained using a rabbit monoclonal anti-p-EGFR (Tyr 1068) antibody, and imaged using a laser-scanning confocal microscope using the same protocols as described above.

## EGFR and p-EGFR single molecule quantification

EGFR-GFP BAC cells were incubated with 10 ng/ml EGF for 30 min, fixed and stained with the rabbit monoclonal anti-p-EGFR (Tyr 1068) antibody as described above. One field of view was sequentially acquired to record bleaching of GFP and fluorescently labelled secondary antibodies. The resulting time series was segmented with MotionTracking as described above, individual objects were tracked for consecutive frames, and the fluorescence intensity of every endosome between two consecutive frames was subtracted to build the ΔIntensity distribution (*Figure 2—figure supplement 3A,C*). The width of distribution is mostly determined by the fluctuations of intensities. However, due to bleaching, the distribution is slightly skewed toward negative values. First, the ΔIntensity was binned. Then the difference between frequencies of negative and positive ΔIntensity of equal absolute values was plotted as function of ΔIntensity (*Figure 2—figure supplement 3B,D*). We named it neg-double-difference function. Since every bin of neg-double-difference function in the vicinity of the first peaks contained ~2500 events, random fluctuations were strongly suppressed and the averaging revealed the discrete structure of bleaching, that is, bleaching of individual molecules. The local amplitude positive maxima correspond to discrete intensity changes when 1, 2, 3, …, n molecules are bleached (see e.g., arrows on peaks at 280, 560, and 800 integral intensity units in *Figure 2—figure supplement 3B*). We estimated that one molecule of GFP and alexa555-antibody corresponds to the integral intensity units at the first peak of the neg-double-difference function (280 and 190 integral intensity units for EGFR-GFP and alexa555, respectively). This method estimated directly the number of EGFR-GFP molecules. The total number of EGFR molecules per endosome was corrected for the ratio of endogenous and BAC EGFR-GFP expressions ($1.29 \pm 0.07$, based on WB quantification). Since the method estimated the number of fluorophores, in the case of antibodies with unknown labelling stoichiometry and epitope accessibility, the result has to be corrected for a scaling factor. The scaling factor of antibody labelling was estimated as 1.9 by comparison of EGFR and p-EGFR distributions at 3 and 5 min of EGF stimulation (10 ng/ml), when most internalized EGFR are still phosphorylated (*Sorkin and Goh, 2009*), with distributions at 30 min.

We used this fluorescence intensity-based method also to estimate the number of EGFR molecules in both clathrin-dependent and independent vesicles. From geometrical calculations, assuming that an uncoated CCV has a diameter of 90 nm and an EGFR dimer has a diameter of ~15 nm and luminal domain of ~10 nm, we estimate that a CCV can contain up to 70 EGFR molecules. However, this calculation does not take into account that vesicles contain multiple types of transmembrane proteins and thus, the value can only be an upper limit. Therefore, we estimated the number of EGFR molecules per diffraction-limited, EEA1-negative vesicle that can be observed following 5 min of 10 ng/ml EGF stimulation. Using this method, we estimated $8.5 \pm 3.5$ molecules/vesicle. Based on this value, we calculated that ~12 vesicles are required to deliver the 102 EGFR molecules/EEA1-positive endosome.

## Fitting of time course kinetics

Time courses were fitted with the sum of two exponential terms: one for growth and one for decline.

$$Ae^{-\frac{t}{\tau_1}} + Be^{-\frac{t}{\tau_2}} \tag{4}$$

The constant of the decline exponent $\tau_2$ was used as an estimation of the decay time of the corresponding process. The fitting of the experimental data was done according to an optimization scheme previously described (*Press et al., 1992*).

### p-EGFR detection in MVBs

To discriminate p-EGFR exposed on the surface of endosomes from p-EGFR sequestered into ILVs, we used a differential detergent solubilisation method as previously described for protease protection assays (*Malerød et al., 2007*). Cells were fixed and permeabilized with saponin 0.1% for 10 min or digitonin 0.001% for 1 min. After permeabilization, cells were washed with PBS and stained with a mouse monoclonal anti-phospho-tyrosine-AlexaFluor 555 antibody (Millipore), a mouse monoclonal anti-LBPA (a gift by J Gruenberg, University of Geneva) antibody, or a mouse monoclonal anti-GFP (Roche, Switzerland) antibody together with a goat anti-mouse-AlexaFluor 555 antibody (Molecular Probes, Invitrogen) to reveal the antigen signal.

Membrane permeabilization with saponin allows access of antibodies both to the cytosol and the luminal content of endosomes, whereas digitonin only to the cytosol. Upon digitonin permeabilization, the staining of LBPA, a marker of ILVs and EGF or EGFR was strongly reduced in comparison with saponin permeabilization (*Figure 2A*), consistent with their localization predominantly within the endosomal lumen. After 30 min of EGF stimulation, the endosomal, but not the plasma membrane EGFR staining was strongly reduced in cells permeabilized with digitonin compared with saponin, probably reflecting the internalization of receptors into ILVs (*Figure 2A,B*). In contrast, the p-EGFR levels were only moderately reduced upon permeabilization with digitonin compared with saponin extraction (*Figure 2A,B*), suggesting that the majority of p-EGFR faced the cytosolic surface of endosomes and was not within ILVs. To measure Shc1 recruitment to endosomes, cells were permeabilized with saponin and stained with a rabbit polyclonal anti-Shc1 antibody (BD Biosciences). Image acquisition, correction, and analysis proceeded as described above.

### dSTORM microscopy

Cells were stimulated for different times with 10 ng/ml EGF and fixed as described above. To detect p-EGFR in endosomes, cells were stained using a rabbit monoclonal anti-p-EGFR (Tyr 1068) antibody. For dSTORM microscopy, the samples were mounted on medium optimized for enhanced switching between fluorescent and non-fluorescent states as previously described (*van de Linde et al., 2011*; *Lampe et al., 2012*). Imaging was performed using a H3 Andor spinning disk microscope with a 100× objective as previously described (*van de Linde et al., 2011*; *Lampe et al., 2012*).

### Calculation of changes in endosome area distributions

First, the binned histograms of endosome area were built with bin widths linear in a logarithmic scale. Then, the histograms were normalized on their integrals, that is, histograms were scaled to have the sum of values in all bins equal to one. Finally, the histogram from the control condition was subtracted from the respective histograms of the different conditions (*Figure 6—figure supplement 2*).

### Mathematical model of p-EGFR propagation through the endosomal network

To describe the time course of the formation of a mean constant amount of p-EGFR per endosome during endocytosis, we postulated a sigmoidal dependency of the dephosphorylation rate on the amount of p-EGFR per endosome. The rationale for this is that if the amount of p-EGFR per endosome is above a critical value, dephosphorylation is significantly increased, whereas if the amount is lower, dephosphorylation is decreased. The delay between EGF stimulation and onset of internalization of p-EGFR into early endosomes is well documented (*Burke et al., 2001*; *Wiley, 2003*). This delay includes EGF binding to receptor (∼3 min), CCV formation (∼1–2 min) and delivery of p-EGFR to early endosomes. In order to keep the model as simple as possible, we described these mechanisms in a coarse grained model by an exponential delay with constant $\delta\tau$. Since the dephosphorylation rate depends on the amount of p-EGFR per endosome, we expanded the mass flux equation usually applied in these cases with an equation that describes the number of endosomes carrying p-EGFR. Our experimental data suggest a significant redistribution of EGFR from the plasma membrane into endosomes even at very low doses of EGF (see *Figure 1E*, green curve. Compare 0.5 with 10 ng/ml). A simple mechanism to explain this is the internalization of ligand-unoccupied EGFR upon EGF stimulation, for example by formation of EGFR oligomers at the plasma membrane (*Ariotti et al., 2010*; *Hofman et al., 2010*). Another possible mechanism includes transient activation of p38 (*Faust et al., 2012*) by EGFR signalling that leads to acceleration of unoccupied receptor internalization

(*Zwang and Yarden, 2006*; *Faust et al., 2012*). Therefore, we modelled the rate of EGFR-positive vesicle formation as $K_v = k_{v0} + k_{v1}\frac{S_p^q}{Q_v^q + S_p^q}$, where $S_p$ is p-EGFR on plasma membrane, $q$ is Hill coefficient, $Q_v$ is a characteristic constant. We considered that the ratio of EGF loaded/unloaded EGFR in the vesicles is equal to the weighted ratio p-EGFR/EGFR on plasma membrane with weight factor $w$. Importantly, the use of this term in the model gave the best description of the time course of total EGFR, but was not essential to explain the p-EGFR dynamics in individual endosomes (data not shown).

$$\frac{dS_m}{dt} = -k_{in} \cdot S_m \cdot c_{EGF}\left(1 - e^{-\left(\frac{t}{\delta\tau}\right)^2}\right) + k_{out} \cdot S_{mp} - K_v \cdot \frac{S_m}{S_m + w \cdot S_{mp}} \cdot S_m + k_{re\_out} \cdot S_{re} \tag{5}$$

$$\frac{dS_{mp}}{dt} = k_{in} \cdot S_m \cdot c_{EGF}\left(1 - e^{-\left(\frac{t}{\delta\tau}\right)^2}\right) - k_{out} \cdot S_{mp} - K_v \cdot \frac{w \cdot S_{mp}}{S_m + w \cdot S_{mp}} \cdot S_{mp} \tag{6}$$

$$\frac{dS_{pe}}{dt} = K_v \cdot \frac{w \cdot S_{mp}}{S_m + w \cdot S_{mp}} \cdot S_{mp} - \left(\beta_1 + \frac{S_{pe}^r}{(Q \cdot N_{pe})^r + S_{pe}^r}(\beta_2 - \beta_1)\right) \cdot S_{pe} \tag{7}$$

$$\frac{dS_e}{dt} = K_v \cdot \frac{S_m}{S_m + w \cdot S_{mp}} \cdot S_m + \left(\beta_1 + \frac{S_{pe}^r}{(Q \cdot N_{pe})^r + S_{pe}^r}(\beta_2 - \beta_1)\right) \cdot S_{pe} - k_{re}S_e - k_{le}S_e \tag{8}$$

$$\frac{dS_{re}}{dt} = k_{re}S_e - k_{re\_out} \cdot S_{re} \tag{9}$$

$$\frac{dN_{pe}}{dt} = \frac{K_v}{s_v}\frac{w \cdot S_{mp}}{S_m + w \cdot S_{mp}} \cdot w \cdot S_{mp} - \rho \cdot N_{pe}^2 + f \cdot N_{pe}, \tag{10}$$

where,

$S_m$ is the amount of non-phosphorylated EGFR on the plasma membrane,

$c_{EGF}$ is the amount of EGF in the extracellular medium,

$S_{mp}$ is the amount of p-EGFR on plasma membrane,

$S_e$ is the non-phosphorylated EGFR on early endosomes,

$S_{pe}$ is the amount of non-phosphorylated EGFR on early endosomes,

$S_{re}$ is the amount of EGFR on recycling endosomes,

$K_v$ is the rate of EGF-stimulated EGFR-positive vesicle formation (see above),

$N_{pe}$ is the number early endosomes with p-EGFR,

$k_{in}$ is the rate of EGF binding to EGFR,

$k_{out}$ is the rate of EGF release from EGFR,

$k_{re}$ is the rate of sorting of EGFR from early to recycling endosomes,

$k_{le}$ is the rate of sorting of EGFR from early to late endosomes,

$k_{re\_out}$ is the rate of delivery of EGFR from recycling endosomes to plasma membrane,

$s_v$ is the mean amount of p-EGFR per endocytic vesicle,

$\beta_1$, $\beta_2$ are minimum and maximum dephosphorylation rates,

$r$ is a Hill coefficient of dephosphorylation rate,

$Q$ is the characteristic amount of p-EGFR at which the dephosphorylation has ½ maximal rate,

$\rho$ is the early endosome homotypic fusion rate (measured as number of events/minute/endosome),

$f$ is the early endosome homotypic fission rate.

*Equation 5* describes the total amount of non-phosphorylated EGFR on the plasma membrane ($S_m$). The first term describes the loss of non-phosphorylated EGFR that becomes phosphorylated upon EGF binding. The second term describes the increase in non-phosphorylated EGFR upon release of EGF from p-EGFR and concomitant dephosphorylation. The third term describes the amount of non-phosphorylated EGFR which is internalized upon EGF-stimulated endocytosis (see above). The last term describes the recycling of EGFR to the plasma membrane.

*Equation 6* describes the total amount of p-EGFR on the plasma membrane ($S_{mp}$). The first term describes the phosphorylation of EGFR upon EGF binding, the second the dephosphorylation following EGF release, and the third its endocytosis.

*Equation 7* describes the amount of p-EGFR in EEA1-positive early endosomes ($S_{pe}$). The first term describes the endocytosis of p-EGFR and the second its dephosphorylation. Note that the equation includes a sigmoidal function $\beta_1 + \frac{S^r_{pe}}{(Q \cdot N_{pe})^r + S^r_{pe}}(\beta_2 - \beta_1)$ for the dephosphorylation rate.

*Equation 8* describes the amount of non-phosphorylated EGFR on early endosomes ($S_e$). The first term describes EGF-stimulated endocytosis of ligand-free EGFR. The second term describes the increase in the amount of non-phosphorylated EGFR through dephosphorylation of p-EGFR. The third and fourth terms describe the sorting of EGFR to recycling and late endosomes, respectively.

*Equation 9* describes the amount of EGFR on recycling endosomes ($S_{re}$). The first term describes the delivery of EGFR from early endosomes and the second its recycling to the plasma membrane.

*Equation 10* describes the number of EEA1-positive early endosomes containing p-EGFR ($N_{pe}$). For simplicity, we considered that the p-EGFR is evenly distributed between endosomes. The first term describes the endocytosis of p-EGFR, the second the homotypic fusion of early endosomes, and the third their homotypic fission.

The model was fitted to the experimental data which included time courses of p-EGFR and EGFR colocalization to EEA1 (*Kalaidzidis et al., 2015*) upon stimulation with four different concentrations (0.5, 1.0, 5.0, and 10.0 ng/ml) of EGF (*Figure 3A,B*). Fitting was performed with FitModel software (*Zeigerer, 2012*) (http://pluk.mpi-cbg.de/projects/fitmodel). Since the amount p-EGFR was measured experimentally in arbitrary FRET intensity units, the modelled amount of p-EGFR was scaled before a comparison with the experimental data. The scaling factor was found by the least square formula

$$scale = \frac{\sum_{i=1}^{N} \frac{d_i \cdot s_i}{\sigma_i^2}}{\sum_{i=1}^{N} \frac{s_i^2}{\sigma_i^2}},$$ where $d_i$, $\sigma_i$; $i = 1 \dots N$ are experimental values and their SEMs; $s_i$ are model predictions

for the respective time points. The model prediction of p-EGFR modulation by reduction of the early endosome homotypic fusion rate is presented on *Figure 3C*. The model and fit parameters are provided in the text format and in the format of FitModel software in the *Source code 1* (Model.zip).

## Knock-down and phenotype characterization in Hela EGFR BAC cells

HeLa EGFR BAC cells were reverse transfected with 5 nM siRNA oligonucleotides per gene using the oligonucleotides given in *Table 2*.

Transfection was carried out using Interferin (Polyplus transfection) together with the selected oligonucleotides following the manufacturer's instructions or treated only with Interferin (mock). 72 hr after transfection total protein extracts were prepared to measure down-regulation of the targeted proteins by western blotting using antibodies previously described for EEA1 and Rabenosyn5 (*Collinet et al., 2010*). To measure the redistribution of EGFR in endosomes, cells were incubated with 10 ng/ml EGF (Invitrogen), fixed, and processed for quantitative microscopy. Image acquisition and analysis were done as described earlier. Measurement of p-EGFR was done using the FRET assay described above. To measure EGFR transport kinetics, cells were incubated in serum-free medium for 1 min with 10 ng/ml EGF (Invitrogen), washed with serum-free medium, and chased for different time points. Cells were then fixed and samples were processed for quantitative microscopy analysis as explained above.

To measure degradation of EGFR, cells were incubated for 1 hr with 10 µg/ml Cyclohexamide before stimulation with 10 ng/ml EGF for different time points. Total protein extracts were prepared and analysed by western blotting using rabbit monoclonal anti-EGFR (Cell Signaling, New England BioLabs, Massachusetts, USA) and mouse anti γ-tubulin (Antibody Facility, MPI-CBG, Germany) antibodies. To measure activated EGFR at the plasma membrane, cells were incubated with 100 ng/ml EGF-AlexaFluor 488 for 10 min on ice to

**Table 2**. List of siRNAs used for down-regulation of endosomal proteins

| Gene name | siRNA library | siRNA ID |
| --- | --- | --- |
| EEA1 | Ambion Silencer | 139147 |
| Rabenosyn5 | Ambion Silencer | 292470 |
| Vps45 | Ambion Silencer | 136363 |
| Hrs | Qiagen | SI00067305 |
| Hrs | Qiagen | SI00288239 |
| Hrs | Qiagen | SI02659650 |
| Vps24 | Invitrogen | 148627 |
| Vps24 | Invitrogen | 148628 |
| Vps24 | Qiagen | SI00760515 |
| Snf8 | Invitrogen | 140086 |
| Snf8 | Qiagen | SI00375641 |
| Snf8 | Qiagen | SI00375648 |

prevent endocytosis, fixed with PFA, stained with a rabbit anti-AlexaFluor 488 antibody fraction (Invitrogen) to enhance the fluorescent signal and imaged as described above.

## Phosphatase siRNA screen

HeLa EGFR BAC cells were reverse transfected with the protocol described above with 5 nM of the oligonucleotides in *Table 3*. After 72 hr, cells were stimulated with 10 ng/ml EGF for 30 min, fixed with PFA, and stained using a rabbit monoclonal anti-p-EGFR (Tyr 1068) antibody as described above. Images were acquired with an automated spinning-disk confocal microscope (OPERA, Evotec Technologies-PerkinElmer) with a 40×/0.9 NA water immersion objective. Settings were adjusted to minimize pixel-intensity saturation and maximize the dynamic range. Around 30 images for each siRNA oligonucleotide were collected. Images with less than three cells were excluded from analysis. Image analysis was performed with MotionTracking as described above.

## MAPK signalling measurements

72 hr after transfection, HeLa EGFR BAC cells were stimulated with 10 ng/ml EGF for different time points. Then, total protein extracts were prepared and analysed by western blotting using rabbit monoclonal anti-phospho-Erk1/2 (Thr202/Tyr204) (Cell Signaling, New England BioLabs) and mouse monoclonal anti-Erk1/2 (Cell Signaling, New England BioLabs) antibodies. For quantification, phospho-Erk1/2 intensity values were first normalized by the total Erk1/2 signal to control for differences in lane loading. For every blot, these values were normalized by the mean intensity amplitude per blot and then scaled by the mean difference between knock-down and mock-treated samples per experiment to account for experimental variability. To measure c-Fos activation, cells were stimulated with 10 ng/ml EGF for 30 min, fixed with PFA, and permeabilized with 0.5% Triton in PBS and 5% BSA as a blocking reagent. Cells were stained with a rabbit monoclonal anti-phospho-c-Fos (Ser32) antibody (Cell Signaling, New England BioLabs) and processed for image analysis. To measure Erk1/2 activation in PC12 cells, cells were stimulated with 100 ng/ml EGF or 50 ng/ml NGF and stained with a rabbit monoclonal anti-phospho-Erk1/2 (Thr202/Tyr204) (Cell Signaling, New England BioLabs) using the same protocol as above. 10 images per condition were acquired using a laser-scanning confocal microscope (Duoscan, Zeiss) with a 40×/1.3 oil objective. Image analysis was carried out as described above.

## Animals

All animal studies were conducted in accordance with German animal welfare legislation and in strict pathogen-free conditions in the animal facility of the Max Planck Institute of Molecular Cell Biology and Genetics, Dresden, Germany. Protocols were approved by the Institutional Animal Welfare Officer (Tierschutzbeauftragter), and necessary licenses were obtained from the regional Ethical Commission for Animal Experimentation of Dresden, Germany (Tierversuchskommission, Landesdirektion Dresden).

## Hepatoblast isolation and culture

Foetal hepatic cells were isolated from C57BL/6JOlaHsd mice, maintained in the animal facility of the MPI-CBG. Pregnancies were dated by the presence of a vaginal plug (embryonic day (E) 0.5). Hepatoblasts were prepared from E14.5 liver as described previously (*Kamiya et al., 1999*). Delta-like 1 (Dlk1) + hepatoblasts were isolated from the E14.5 hepatic cells as described previously with minor modifications (*Tanimizu et al., 2003*). Briefly, cells were blocked with an anti-mouse CD16/32 (BD Biosciences) and stained with a FITC-conjugated anti-Dlk1 antibody (MBL International, Massachusetts, USA) followed by anti-FITC Microbeads (Miltenyi Biotec GmbH, Germany). The labelled cells were separated using a MACS Cell Separation Column (Miltenyi Biotec). Dlk1+ cells were resuspended in DMEM (PAA Laboratories GmbH, Austria) containing 5% FBS, 2 mM L-glutamine (PAA Laboratories GmbH), 100 μM MEM Non-Essential Amino Acids (PAA Laboratories GmbH), 0.1 μM dexamethasone (Sigma–Aldrich), 100 Units/ml penicillin (PAA Laboratories GmbH), 100 μg/ml streptomycin (PAA Laboratories GmbH), and 4% BD Matrigel Basement Membrane Matrix (BD Biosciences), and seeded on a μ-slide 8-well (Ibidi GmbH, Germany) coated with fibronectin

**Table 3.** List of genes for PTP siRNA screen

| Gene symbol | Gene ID | siRNa ID | Sequence 5′–3′ |
| --- | --- | --- | --- |
| PTPN13 | 5783 | 5783-HSS108838 | UCACAUUUCUGAACCAACUAGACAA |
| PTPN13 | 5783 | 5783-HSS184076 | CAUCAGACUCUAAGCAACAUGGUAU |
| PTPN13 | 5783 | 5783-HSS184077 | CCAUUGAGGGUAAUCUCCAGCUAUU |
| PTPN13 | 5783 | 5783-NM_080683.1_1459 | GAAACACCCUUUGAAGGCAACUUAA |
| PTPRK | 5796 | 5796-HSS108869 | CCCAUCCAAGUGGAAUGUAUGUCUU |
| PTPRK | 5796 | 5796-HSS108870 | GGUCAUUCUUGAAACUGAUACUUCA |
| PTPRK | 5796 | 5796-HSS184093 | CCGCGCAAAGGAUACAACAUCUAUU |
| PTPRK | 5796 | 5796-NM_002844.2_975 | CCGCUUCCUUCAGAUUGCAAGAAGU |
| PTPRA | 5786 | 5786-HSS108844 | CCAGUUCACGGAUGCCAGAACAGAA |
| PTPRA | 5786 | 5786-HSS108845 | GCAUUCUCAGAUUAUGCCAACUUCA |
| PTPRA | 5786 | 5786-HSS108846 | GGCACCAACAUUCAGCCCAAAUAUA |
| PTPRA | 5786 | 5786-NM_080841.2_1383 | CGCCUCAUCACUCAGUUCCACUUUA |
| PTPRR | 5801 | 5801-HSS108880 | AGUUGAGGUUCUGGUUAUCAGUGUA |
| PTPN9 | 5780 | 5780-HSS108830 | CCCUCAUUGACUUCUUGAGAGUGGU |
| PTPRR | 5801 | 5801-HSS108882 | GGUACACCUCAUGGCCUGAUCACAA |
| PTPN9 | 5780 | 5780-HSS108831 | ACCUCAUGAGGAACCUCUUCGUUCU |
| PTPRR | 5801 | 5801-HSS184097 | CAAGAGAGAAGAGGGUCCAACGUAU |
| PTPN9 | 5780 | 5780-HSS184065 | CGCUGUCUUGGAAUGUGGCUGUCAA |
| PTPRR | 5801 | 5801-NM_130846.1_1022 | CAGUGGCAAGGAGAAAGCCUUCAUU |
| PTPN9 | 5780 | 5780-NM_002833.2_1369 | CAUCCAAGAGUUGGUGGACUAUGUU |
| PTPN2 | 5771 | 5771-HSS108817 | GGAAGACUUAUCUCCUGCCUUUGAU |
| PTPN2 | 5771 | 5771-HSS108818 | GAGCGGGAGUUCGAAGAGUUGGAUA |
| PTPN2 | 5771 | 5771-HSS184039 | GAGAUUCUCAUACAUGGCUAUAAUA |
| PTPN2 | 5771 | 5771-NM_002828.2_1178 | CCGAUGUACAGGACUUUCCUCUAAA |
| DUSP2 | 1844 | 1844-HSS140936 | GCUCUGCCACCAUCUGUCUGGCAUA |
| PTPN3 | 5774 | 5774-HSS108820 | GGCGUGGUACAGACCUUUAAAGUUA |
| PTPRE | 5791 | 5791-HSS108853 | UCUGGGAAUGGAAAUCCCACACUAU |
| DUSP2 | 1844 | 1844-HSS140937 | GCUGCUGUCCCGAUCUGUGCUCUGA |
| PTPN3 | 5774 | 5774-HSS108821 | GAGCUGUCCGCUCAUUUGCUGACUU |
| PTPRE | 5791 | 5791-HSS108854 | ACGAGACUUUCUGGUCACUCUCAAU |
| DUSP2 | 1844 | 1844-HSS140938 | GGCAUCACAGCCGUCCUCAACGUGU |
| PTPN3 | 5774 | 5774-HSS108822 | CCACCCGGGUAUUAUUGCAGGGAAA |
| PTPRE | 5791 | 5791-HSS108855 | GGAACAGUAUGAAUUCUGCUACAAA |
| DUSP2 | 1844 | 1844-NM_004418.3_925 | UGGACGAGGCCUUUGACUUCGUUAA |
| PTPN3 | 5774 | 5774-NM_002829.2_621 | CAAUCAGAAGCAGAAUCCUGCUAUA |
| PTPRE | 5791 | 5791-NM_130435.2_1499 | GAGCAGGAUAAAUGCUACCAGUAUU |
| PTPRF | 5792 | 5792-HSS108856 | CCCAUCAUCCAAGACGUCAUGCUAG |
| PTPRF | 5792 | 5792-HSS108858 | GGACAGCAGUUCACGUGGGAGAAUU |
| PTPRF | 5792 | 5792-HSS184088 | CAGCUGUGCCCUUUAAGAUUCUGUA |
| PTPRF | 5792 | 5792-NM_130440.2_6013 | CAGCUUUGACCACUAUGCAACGUAA |
| PTP4A2 | 8073 | 8073-HSS140957 | GAUAACUCACAACCCUACCAAUGCU |
| DUSP6 | 1848 | 1848-HSS176270 | GAGAGCAGCAGCGACUGGAACGAGA |

*Table 3. Continued on next page*

*Table 3. Continued*

| Gene symbol | Gene ID | siRNa ID | Sequence 5′–3′ |
| --- | --- | --- | --- |
| PTP4A2 | 8073 | 8073-HSS140958 | GCGUUCAAUUCCAAACAGCUGCUUU |
| DUSP6 | 1848 | 1848-HSS176271 | UGGCAUUAGCCGCUCAGUCACUGUG |
| PTP4A2 | 8073 | 8073-HSS188476 | GGUUCGAGUUUGUGAUGCUACAUAU |
| DUSP6 | 1848 | 1848-HSS176272 | UGGCUUACCUUAUGCAGAAGCUCAA |
| PTP4A2 | 8073 | 8073-NM_080392.2_1123 | UCGAGUUUGUGAUGCUACAUAUGAU |
| DUSP6 | 1848 | 1848-NM_022652.2_1097 | CAUGUGACAACAGGGUUCCAGCACA |
| PTPRM | 5797 | 5797-HSS108871 | CCGAGUGAGGCUGCAGACAAUAGAA |
| PTP4A3 | 11156 | 11156-NM_007079.2_423 | UCAGCACCUUCAUUGAGGACCUGAA |
| PTPN18 | 26469 | 26469-HSS120076 | GCUGCCUUAUGAUCAGACGCGAGUA |
| PTPN14 | 5784 | 5784-HSS108841 | UCAUGGGAAUGAAGAAGCCUUGUAU |
| PTPRM | 5797 | 5797-HSS108872 | CAGGCUCUGGUUACAGGGCAUUGAU |
| PTP4A3 | 11156 | 11156-NM_007079.2_460 | UACCACUGUGGUGCGUGUGUGUGAA |
| PTPN18 | 26469 | 26469-HSS120077 | UCGAGAGAUAGAGAAUGGGCGGAAA |
| PTPN14 | 5784 | 5784-HSS108843 | GCCGCUGAUGUUGGCAGCAUUGAAU |
| PTPRM | 5797 | 5797-HSS108873 | CCCGACGCUUCAUUGCUUCAUUUAA |
| PTP4A3 | 11156 | 11156-NM_007079.2_473 | CGUGUGUGUGAAGUGACCUAUGACA |
| PTPN18 | 26469 | 26469-HSS120078 | CCCACCUGACUUCAGUCUCUUUGAU |
| PTPN14 | 5784 | 5784-HSS184078 | GAUAUCAGUAUUACCUGCAAGUCAA |
| PTPRM | 5797 | 5797-NM_002845.3_1217 | CCGACGCUUCAUUGCUUCAUUUAAU |
| PTP4A3 | 11156 | 11156-NM_007079.2_678 | CCAUCAACAGCAAGCAGCUCACCUA |
| PTPN18 | 26469 | 26469-NM_014369.2_835 | UCAGUCUCUUUGAUGUGGUCCUUAA |
| PTPN14 | 5784 | 5784-NM_005401.3_3394 | CACGAAGUUUCGAACGGAUUCUGUU |
| PTPN1 | 5770 | 5770-HSS108816 | GAGUGAUGGAGAAAGGUUCGUUAAA |
| PTPN1 | 5770 | 5770-HSS184025 | CAUGAAGCCAGUGACUUCCCAUGUA |
| PTPN1 | 5770 | 5770-HSS184026 | CGAGAGAUCUUACAUUUCCACUAUA |
| PTPN1 | 5770 | 5770-NM_002827.2_507 | CAGAGUGAUGGAGAAAGGUUCGUUA |
| PTPRJ | 5795 | 5795-HSS108867 | GCGACUUCAUAUGUAUUCUCCAUCA |
| PTPRJ | 5795 | 5795-HSS184091 | CGGGUUCUUCUUGAAAGCAUUGGAA |
| PTPRJ | 5795 | 5795-HSS184092 | GAGCAGCCAUGAUGCAGAAUCAUUU |
| PTPRJ | 5795 | 5795-NM_002843.3_1838 | CGGGUAGAAAUAACCACCAACCAAA |
| PTPN12 | 5782 | 5782-HSS108835 | GCCACAGGAAUUAAGUUCAGAUCUA |
| PTP4A1 | 7803 | 7803-HSS111748 | GCAACUUCGUAUUUGGAGAAGUAU |
| PTPN12 | 5782 | 5782-HSS108836 | GCCUCUUGAUGAGAAAGGACAUGUA |
| PTP4A1 | 7803 | 7803-HSS111749 | UCAAAGAUUCCAACGGUCAUAGAAA |
| PTPN12 | 5782 | 5782-HSS108837 | UCUGAUGGUGCUGUGACCCAGAAUA |
| PTP4A1 | 7803 | 7803-HSS111750 | CCAACCAAUGCGACCUUAAACAAAU |
| PTPN12 | 5782 | 5782-NM_002835.2_554 | CAGGACACUCUUACUUGAAUUUCAA |
| PTP4A1 | 7803 | 7803-NM_003463.3_1382 | AACCAGAUUGUUGAUGACUGGUUAA |
| PTPN11 | 5781 | 5781-HSS108834 | ACAUGGAACAUCACGGGCAAUUAAA |
| PTPN11 | 5781 | 5781-HSS184068 | CAGACAGAAGCACAGUACCGAUUUA |
| PTPN11 | 5781 | 5781-HSS184069 | GAAAGGGCACGAAUAUACAAAUAUU |
| PTPN11 | 5781 | 5781-NM_002834.3_5519 | CAGGAUGCCUUUGUUAGGAUCUGUA |

(Sigma–Aldrich, Germany). To measure Erk1/2 activation, cells were starved for 24 hr before stimulation with either 10 ng/ml EGF or HGF (R&D systems, Minnesota, USA). Total cell lysates were prepared and analysed using the same protocol and antibodies described above.

## EEA1 staining after growth factor stimulation in PC12 cells or hepatoblasts

We used a clone of PC12 cells, PC12 Nsc-1 (Cellomics Inc., Maryland, USA) cells, due to their increased growth rate and decreased cell clumping, which facilitate imaging experiments (*Hahn et al., 2009*). Cells were grown following the manufacturer's instructions. PC12 cells were starved for 36 hr before stimulation either with 100 ng/ml EGF (Invitrogen) or 50 ng/ml NGF (R&D Systems) for 30 min. E14.5 Dlk1+ hepatoblasts were starved for 24 hr before stimulation with either 10 ng/ml EGF or HGF (R&D systems). Then, cells were fixed with 3% para-formaldehyde and stained with a mouse monoclonal anti-EEA1 (BD Biosciences Pharmingen). A fluorescently conjugated goat anti-mouse-AlexaFluor 555 secondary antibody (Molecular Probes, Invitrogen) revealed the antigen signal. Image acquisition and image analysis were performed as described above.

## Triple knock-down and phenotype characterization in PC12 Nsc-1 cells

PC12 Nsc-1 cells were electroporated with 100 nM Stealth Select siRNA oligonucleotides (Invitrogen) (EEA1: 5′—GAA AGC AGC UCA ACU UGC UAC UGA A—3′, 3′—UUC AGU AGC AAG UUG AGC UGC UUU C—5′; Rabenosyn5: 5′—GGG CCU CAC ACU GAU CUU GCC UAU U—3′, 3′—AAU AGG CAA GAU CAG UGU GAG GCC C—5′; Vps45: 5′—GAC CCG GCA UGA AGG UAC UUC UCA U—3′, 3′—AUG AGA AGU ACC UUC AUG CCG GGU C—5′; Syntaxin-13: 5′—CCA GGU GAU CUU GAU UGA UAG CAU A—3′, 3′—UAU GCU AUC AAU CAG AUC ACC UUG G—5′; Syntaxin-6: 5′—GGA UGC UGG AGU GAC GGA UCG AUA U—3′, 3′—AUA UCG AUC CGU CAC UCC AGC AUC C—5′) or electroporated without siRNAs (Mock) using the Amaxa Cell line Nucleofector Kit V (Lonza, Switzerland) following the manufacturer's instructions. 36 hr after electroporation, cells were placed in serum-free medium. 72 hr after electroporation total protein extracts were prepared to measure down-regulation of the targeted proteins by western blotting with a mouse monoclonal anti-Syntaxin-13 (Synaptic Sytems, Germany) or a mouse monoclonal anti-Syntaxin-6 (Transduction Laboratories, BD Biosciences) antibody. To measure EGF transport, cells were stimulated with 100 ng/ml EGF-Alexafluor 555 (Molecular Probes, Invitrogen) for 1 min, washed with serum-free medium, and chased for different times. Then, cells were fixed and processed for microscopy as described above.

## PC12 Nsc-1 differentiation-proliferation assay

Cells were starved for 36 hr and then stimulated in serum-free medium with 100 ng/ml EGF (Invitrogen) or 50 ng/ml NGF (R&D Systems) for 24 hr at 37°C and 5% $CO_2$. During the last 3 hr, 5-ethynyl-2′ –deoxyuridine (EdU) was added at a final concentration of 10 µM. Then, cells were fixed and stained with Click-iT AlexaFluor 647 Azide (Molecular Probes, Invitrogen) following the manufacturer's instructions. Afterwards, cells were stained with a mouse monoclonal anti-β-III tubulin antibody (Chemicon International, Millipore) and a fluorescently conjugated goat anti-mouse-AlexaFluor 555 (Molecular Probes, Invitrogen) to reveal the antigen signal. Nuclei were stained with DAPI. 20 images per condition were acquired using a laser-scanning confocal microscope (Duoscan, Zeiss) with a 20×/0.8 objective. Image processing was carried out as described above. Images were inspected manually for process formation; cells with processes were defined as those having thin β-III tubulin-positive processes longer than 1 µm. The β-III tubulin expression was measured by total immunofluorescence intensity normalized by the frame area covered by cells to account for frame-to-frame variability in cell number.

## Acknowledgements

We acknowledge T Galvez, G O'Sullivan, MP McShane, S Eaton, J Rink, J Howard, and P Bastiaens for discussions and comments on the manuscript. We thank Anja Zeigerer and Sarah Seifert for help in the experiments with primary mouse hepatocytes. We acknowledge the MPI-CBG services and facilities, in particular J Peychl for the management of the Light Microscopy Facility, C Möbius (HT-TDS) for

assistance in automated image acquisition, and I Poser for generation of the stable HeLa cell lines. This work was financially supported by the Virtual Liver initiative (www.virtual-liver.de) funded by the German Federal Ministry of Research and Education (BMBF), the Max Planck Society (MPG), and the German Research Foundation (DFG). RV was supported by a grant from the Gottlieb Daimler und Karl Benz Stiftung.

## Additional information

### Funding

| Funder | Grant reference number | Author |
|---|---|---|
| Bundesministerium für Bildung und Forschung | Virtual Liver Initiative | Roberto Villaseñor, Hidenori Nonaka, Perla Del Conte-Zerial, Yannis Kalaidzidis, Marino Zerial |
| Max-Planck-Gesellschaft | | Roberto Villaseñor, Hidenori Nonaka, Perla Del Conte-Zerial, Yannis Kalaidzidis, Marino Zerial |
| Deutsche Forschungsgemeinschaft | | Roberto Villaseñor, Hidenori Nonaka, Perla Del Conte-Zerial, Yannis Kalaidzidis, Marino Zerial |
| Daimler und Benz Stiftung | | Roberto Villaseñor |

The funders had no role in study design, data collection and interpretation, or the decision to submit the work for publication.

### Author contributions

RV, Conception and design, Acquisition of data, Analysis and interpretation of data, Drafting or revising the article; HN, Established procedures for the isolation, culture and staining conditions of the embryonic hepatoblasts, Contributed unpublished essential data or reagents; PDC-Z, Developed the mathematical model and simulations, Analysis and interpretation of data; YK, Developed the FRET correction algorithm, Developed method to estimate the number of fluorescent molecules in light microscopy images, Supervised the image analysis, Developed the mathematical model and simulations, Conception and design, Analysis and interpretation of data, Drafting or revising the article; MZ, Directed Study, Conception and design, Drafting or revising the article

### Ethics

Animal experimentation: All animal studies were conducted in accordance with German animal welfare legislation and in strict pathogen-free conditions in the animal facility of the Max Planck Institute of Molecular Cell Biology and Genetics, Dresden, Germany. Protocols were approved by the Institutional Animal Welfare Officer (Tierschutzbeauftragter) under the license Anzeige der Tötung von Tieren zu wissenschaftlichen Zwecken AZ: 24-9168.24-9/2009-1 (valid from 2009 until 31.12.2012) and AZ: 24-9168.24-9/2012-1 (valid from 30.4.2012 through 30.4.2015), obtained from the regional Ethical Commission for Animal Experimentation of Dresden, Germany (Tierversuchskommission, Landesdirektion Dresden).

## Additional files

### Supplementary file

• Source code 1. System of differential equations for the mathematical model of p-EGFR endocytosis. The ZIP file contains the system of differential equations of the model with non-linear dephosporylation term and experimental data which were used to fit the model (see *Figure 3* of main text). The data are provided in the FitModel format (http://pluk.mpi-cbg.de/projects/fitmodel) and as a simple text.

## Major dataset

The following previously published dataset was used:

| Author(s) | Year | Dataset title | Dataset ID and/or URL | Database, license, and accessibility information |
|---|---|---|---|---|
| Lu C, Mi LZ, Grey MJ, Zhu J, Graef E, Yokoyama S, Springer TA | 2010 | The Extracellular and Transmembrane Domain Interfaces in Epidermal Growth Factor Receptor Signaling | 3NJP | Publicly available at RCSB Protein Data Bank (http://www.rcsb.org). |

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
