## [Decision Letter]

Thank you for sending your work entitled “Regulation of EGFR signal transduction
by analogue-to-digital conversion in endosomes”for consideration at
*eLife*. Your article has been favorably evaluated by Randy Schekman
(Senior editor) and 3 reviewers, one of whom (Suzanne Pfeffer) is a member of our Board
of Reviewing Editors.

The reviewers thought that the manuscript would be improved by clarification of the text
in three areas (no additional experiments are needed).

1) Please report EGFR molecule number per endosomal volume and clarify that if there are
100 EGFR molecules per endosome, this is the result of roughly how many coated vesicles
fusing to bring early endosomes to that number?

2) Please clarify your logic that EGFR kinase activity is regulating phosphatase
activity that, in turn, is regulating endosome pEGFR levels. It is difficult to envision
how regulating something like local enzymatic activities could give rise to such precise
levels of p-EGFR in endosomes. Instead, it seems more compatible with a process whereby
there is a single scaffold (or similar protein assembly) per endosome that binds p-EGFR
and protects it from dephosphorylation. This is supported by your observation that at
low concentrations of EGF, blocking kinase activity does not change the total levels of
p-EGFR, which would be expected if it controlled phosphatase activity, but does reduce
the number of endosomes in which p-EGFR is found. This argues for a role of kinase
activity in the segregation process by which a set number of p-EGFR molecules are
associated with an endosome, not phosphatase activity.

Thus, increasing the levels of p-EGFR by using high concentrations of EGF would reduce
the fraction associated with the scaffolds, resulting in an increase in fractional
dephosphorylation. This hypothesis does correlate higher total EGFR kinase activity to
higher p-EGFR dephosphorylation, but only indirectly. As a consequence of this
correlation, one could develop a good descriptive model that functionally links the two
processes and, for example, could give rise to the prediction that the total amount of
pEGFR is dependent on the fusion/fission rate of endosomes. If p-EGFR flowed into
endosomes too rapidly such that there was insufficient time to assemble the protective
scaffold, their net rate of dephosphorylation would be higher, thus give rise to a
positive correlation between EGFR kinase activity and dephosphorylation, and the
predicted negative relationship between fusion rates and p-EGFR content per
endosome.

(Please discuss this in the text to clarify your thinking for the reader. It would be
best to provide a few possible models and state that more work is needed to distinguish
between them.)

3) The paper would benefit from a one page summary discussing overall molecular
mechanisms for what they observe for non-EGFR experts. This would include the mechanisms
by which different growth factors influence fission and fusion of endosomes, how Hrs and
Shp2 relate to p-EGFR in different cell types stimulated with different growth factors
(or large versus small endosomes).

---

## [Author Response]

*The reviewers thought that the manuscript would be improved by clarification of
the text in three areas (no additional experiments are needed)*.

*1) Please report EGFR molecule number per endosomal volume and clarify that if
there are 100 EGFR molecules per endosome*, *this is the result of
roughly how many coated vesicles fusing to bring early endosomes to that
number?*

We estimated the number of receptors per µm^3^ of endosomal volume
(“we estimated an average of 102±38 and 76±29 (Mean±SEM)
molecules of EGFR and p-EGFR per endosome 30 minutes after EGF (10 ng/ml)
internalization (Figure 2—figure supplement 3), corresponding to 707±265 and 527±202 molecules per
μm^3^ of endosomal volume (apparent, assessed by light microscopy),
respectively”). To estimate the number of CCVs required to bring 100 EGFR per
endosome, we used both geometrical calculations and the intensity based method described
in the Methods.

*2) Please clarify your logic that EGFR kinase activity is regulating phosphatase
activity that, in turn, is regulating endosome pEGFR levels. It is difficult to
envision how regulating something like local enzymatic activities could give rise to
such precise levels of p-EGFR in endosomes. Instead, it seems more compatible with a
process whereby there is a single scaffold (or similar protein assembly) per endosome
that binds p-EGFR and protects it from dephosphorylation. This is supported by your
observation that at low concentrations of EGF, blocking kinase activity does not
change the total levels of p-EGFR, which would be expected if it controlled
phosphatase activity, but does reduce the number of endosomes in which p-EGFR is
found. This argues for a role of kinase activity in the segregation process by which
a set number of p-EGFR molecules are associated with an endosome, not phosphatase
activity*.

*Thus, increasing the levels of p-EGFR by using high concentrations of EGF would
reduce the fraction associated with the scaffolds, resulting in an increase in
fractional dephosphorylation. This hypothesis does correlate higher total EGFR kinase
activity to higher p-EGFR dephosphorylation, but only indirectly. As a consequence of
this correlation, one could develop a good descriptive model that functionally links
the two processes and, for example, could give rise to the prediction that the total
amount of pEGFR is dependent on the fusion/fission rate of endosomes. If p-EGFR
flowed into endosomes too rapidly such that there was insufficient time to assemble
the protective scaffold, their net rate of dephosphorylation would be higher, thus
give rise to a positive correlation between EGFR kinase activity and
dephosphorylation, and the predicted negative relationship between fusion rates and
p-EGFR content per endosome*.

*(Please discuss this in the text to clarify your thinking for the reader. It
would be best to provide a few possible models and state that more work is needed to
distinguish between them*.*)*

We took the reviewers’ suggestion and added a scholarly discussion on the
molecular mechanisms that could be responsible for the formation of p-EGFR quanta at the
end of the Discussion, considering which experimental data are consistent or
inconsistent with each mechanism. We thank the reviewers for their proposal as we
believe this discussion is now clearer and very stimulating. We acknowledge that this is
only possible with *eLife* given that the printed journals normally
demand to trim the text.

*3) The paper would benefit from a one page summary discussing overall molecular
mechanisms for what they observe for non-EGFR experts. This would include the
mechanisms by which different growth factors influence fission and fusion of
endosomes, how Hrs and Shp2 relate to p-EGFR in different cell types stimulated with
different growth factors (or large versus small endosomes)*.

The explanation from point 2 is written as a summary that details how Hrs and SHP2 could
also regulate the formation of quanta for other RTKs. We also report the evidence for
general molecular mechanisms whereby different RTKs could regulate fusion/fission.